# Nonthermal acceleration of protein hydration by sub-terahertz irradiation

Jun-ichi Sugiyama [1,7], Yuji Tokunaga [2,7], Mafumi Hishida[3,6,7], Masahito Tanaka [4], Koh Takeuchi [2], Daisuke Satoh [4] & Masahiko Imashimizu [5] ✉

The collective intermolecular dynamics of protein and water molecules, which overlap in the sub-terahertz (THz) frequency region, are relevant for expressing protein functions but remain largely unknown. This study used dielectric relaxation (DR) measurements to investigate how externally applied sub-THz electromagnetic fields perturb the rapid collective dynamics and influence the considerably slower chemical processes in protein–water systems. We analyzed an aqueous lysozyme solution, whose hydration is not thermally equilibrated. By detecting time-lapse differences in microwave DR, we demonstrated that sub-THz irradiation gradually decreases the dielectric permittivity of the lysozyme solution by reducing the orientational polarization of water molecules. Comprehensive analysis combining THz and nuclear magnetic resonance spectroscopies suggested that the gradual decrease in the dielectric permittivity is not induced by heating but is due to a slow shift toward the hydrophobic hydration structure in lysozyme. Our findings can be used to investigate hydration-mediated protein functions based on sub-THz irradiation.

Proteins must be surrounded by hydrating water molecules to be able to express their functions, such as highly accurate enzymatic reactions[1–3]. The characteristics of water dynamics in the hydration layers of biomolecules have been extensively investigated by spectroscopic approaches and molecular dynamics simulations[4–6]. Sophisticated approaches, including optical Kerr-effect, extraordinary acoustic Raman, and terahertz (THz) spectroscopies, have been developed to analyze biomacromolecular dynamics in the THz frequency region[7–12]. The development of these techniques has revealed that hydrated proteins under ambient conditions are characterized by multicomponents of relaxational and vibrational dynamics at sub-THz frequencies, which have been assigned as the coupled motion of water molecules and protein surface[13–15]. This prompted us to hypothesize that the intense sub-THz radiation can collectively excite biologically relevant coupled protein–water dynamics occurring at the same frequency. Indeed, several studies using 0.1–0.5 THz light sources have shown that irradiation at these frequencies locally increases the electron density in the α-helix of the hydrated lysozyme crystal[16], significantly influences actin polymerization[17], gene expression[18], and cell membrane permeability[19]. Using nuclear magnetic resonance (NMR) spectroscopy, we have previously demonstrated that incident 0.1 THz radiation energy induces structural dynamic changes of ubiquitin, which are opposite to those induced by increased temperature[20]. This result suggests that the collective dynamics include components that

[1]Nanomaterials Research Institute, National Institute of Advanced Industrial Science and Technology, Tsukuba 305-8565, Japan. [2]Graduate School of Pharmaceutical Sciences, The University of Tokyo, Hongo, Bunkyo, Tokyo 113-0033, Japan. [3]Department of Chemistry, Faculty of Pure and Applied Sciences, University of Tsukuba, Tsukuba, Ibaraki 305-8571, Japan. [4]Research Institute for Measurement and Analytical Instrumentation, National Institute of Advanced Industrial Science and Technology, Tsukuba 305-8568, Japan. [5]Cellular and Molecular Biotechnology Research Institute, National Institute of Advanced Industrial Science and Technology, Tsukuba 305-8565, Japan. [6]Present address: Department of Chemistry, Faculty of Science, Tokyo University of Science, 1-3 Kagurazaka, Shinjuku, Tokyo 162–8601, Japan. [7]These authors contributed equally: Jun-ichi Sugiyama, Yuji Tokunaga, Mafumi Hishida. ✉e-mail: m.imashimizu@aist.go.jp

are nonthermally influenced by sub-THz radiation. However, it remains unclear whether the mechanisms behind these phenomena are the same, and if they are, how the radiation energy can be retained in the specific interactions among biomolecules and water molecules.

Using dielectric relaxation (DR) measurement of an aqueous protein solution, the sub-THz-radiation-induced effects on the coupled protein–water dynamics can be evaluated as changes in the rotational freedom of water dipoles and the collective relaxation that are affected by hydrogen-bond (H-bond) networks[21,22]. Under ambient conditions, multiple relaxation components of protein-hydrated and bulk water are present in the microwave frequency region[23]. The relaxation components reflecting the orientational polarization of heterogenous hydration water are termed $\delta$ relaxation and are observed in the frequency range of 0.1–10 GHz for globular protein solutes[24–27]. Therefore, the microwave DR spectroscopy provides direct information on molecular motions responsible for protein hydration as orientational polarization to external electromagnetic (EM) fields. Conventional DR measurement requires a > 5.0 mm thick liquid sample[28]; however, the penetration depth of sub-THz radiation in liquid water is <1.0 mm. Therefore, measuring the microwave DR under sub-THz irradiation is technically challenging.

In this study, we develop a highly sensitive technique for detecting time-lapse differences in microwave DR coupled with sub-THz irradiation, by modulating a standing wave signal that is generated at the sample cell boundaries. We find that the dielectric permittivity of an aqueous globular protein solution, which is not thermally equilibrated, gradually decreases upon intense sub-THz irradiation. Our comprehensive analysis combining microwave DR and THz and NMR spectroscopies suggests that the gradual decrease in the dielectric permittivity originates from a slow change into a more hydrophobic hydration structure of the protein. During irradiation, we can also detect the selective perturbation to the fast water dynamics generated by interacting with the protein.

## Results

### Time-lapse DR measurement coupled with sub-THz irradiation

We developed a reflection method that facilitated time-lapse dielectric measurements of a liquid sample in the microwave frequency range subjected to sub-THz irradiation (Supplementary Fig. 1a). In particular, the liquid sample was exposed to a low-intensity external EM field generated by a vector network analyzer (VNA), and the dielectric response to the field was detected in the presence or absence of intense 0.1 THz exposure from the opposite side of the VNA-generated field through the sample. The sample path length ($l$), defined as the distance between a coaxial probe and the interface of a polydimethylsiloxane (PDMS) container, was fixed at 1.0 mm. As described in Supplementary Notes and Supplementary Figs. 1–3, we utilized the multiple reflections generated specifically by this short $l$, resulting in a standing wave signal, for highly sensitive detection of changes in complex dielectric permittivity.

For sub-THz excitation, we directed intense 0.1 THz pulses of 16 mW/cm$^2$ average power density toward the bottom of the sample by constructing an optical setup using sub-THz-klystron (Supplementary Fig. 1a). The average electric field strength at the irradiation surface was estimated to be ~0.15 kV/m, and no interference of the 0.1 THz field with the VNA-generated field was detected (see Methods). We then measured the elevation in the volume-averaged temperature for the sub-THz-irradiated sample by immersing a resistance temperature detector (Pt100) in the sample solution. Because of the absorption of the sub-THz radiation by water, the sample temperature was gradually increased by ~4 °C during 10 min of irradiation. Similarly, for the high or low temperature control experiments (HTC or LTC), we increased the sample temperature slightly higher than that of sub-THz irradiation (by ~6 °C) or decreased it by ~4 °C through a Peltier stage inserted under the PDMS container (Fig. 1a). Following irradiation or heating/

cooling for 10 min, the slow changes in the dielectric response to each perturbation were further monitored for 30 min (Fig. 1a). The sample temperature returned to the room temperature of 24 °C approximately 20 min after the perturbation. For the general control experiment (GC) performed at a constant temperature, the same experimental procedure was performed using the same sub-THz klystron setup without supplying the sub-THz radiation.

### DR analysis to detect sub-THz irradiation effects

We employed an aqueous solution of the enzyme lysozyme as the sample. Lysozyme is one of the most widely physicochemically studied proteins. The hydration properties of lysozyme have been extensively investigated using dielectric and vibrational spectroscopic approaches in experiments[15,26,27,29–31] and simulations[32–35]. To prepare a lysozyme sample, crystalline lysozyme powder of 30 mg or 100 mg was dissolved in 1 mL pure water (2.9 or 9.1 wt%) for ~2 h prior to measurements.

The complex permittivity of the lysozyme sample was determined by analyzing the difference in impedance at the interface of an open-ended probe, i.e., it was determined by the reflection method using Open, Short, and Standard calibrations of the probe interface (Fig. 1b). In contrast to the standard reflection method, we used the same lysozyme sample before and after 0.1 THz irradiation for the Standard and Unknown measurements, respectively, facilitating precise time-lapse dielectric measurements of the same sample. This procedure minimizes the spectral changes caused by factors other than 0.1 THz irradiation, such as slight differences in the configuration of the coaxial transmission line and in the sample conductivity and volume between the Standard and Unknown measurements, and permits accurate data fitting and extrapolation, which are essential for the subsequent data analysis.

The dielectric response of aqueous lysozyme solution in MHz–GHz frequency regions includes protein-derived relaxation at ~10 MHz ($\beta$) and two hydration water-derived relaxations at ~0.1 and ~4 GHz ($\delta_1$ and $\delta_2$), respectively, at room temperature[26,27]. In addition, slow and fast relaxations from bulk water appear at ~20 GHz and 150–600 GHz ($\gamma_1$ and $\gamma_2$, the exact frequency is debatable), respectively[36–41]. Therefore, the distribution of multiple relaxations involving Debye processes can be expressed as Eq. (1) (Fig. 1c; Nyquist plot for $\varepsilon^*(\omega)$).

$$\varepsilon^*(\omega) = \varepsilon(\infty)_{\text{relax}} + \frac{\Delta\varepsilon_\beta}{1+j\omega\tau_\beta} + \frac{\Delta\varepsilon_{\delta1}}{1+j\omega\tau_{\delta1}} + \frac{\Delta\varepsilon_{\delta2}}{1+j\omega\tau_{\delta2}} + \frac{\Delta\varepsilon_{\gamma1}}{1+j\omega\tau_{\gamma1}} + \frac{\Delta\varepsilon_{\gamma2}}{1+j\omega\tau_{\gamma2}},$$
(1)

where $\Delta\varepsilon$ is the strength of each relaxation, $\tau$ is the relaxation time, and $\varepsilon(\infty)_{\text{relax}}$ is the apparent high-frequency limit in the Debye-type relaxation comprising all vibrational components ($\Delta\varepsilon_{\text{vib}}$) of the higher frequency regions and the high-frequency limit:

$$\varepsilon(\infty)_{\text{relax}} = \sum \Delta\varepsilon_{\text{vib}} + \varepsilon(\infty).$$
(2)

Taking the limit of $\omega \to 0$, Eq. (3) gives static dielectric constant:

$$\varepsilon(s) = \Delta\varepsilon_\beta + \Delta\varepsilon_{\delta1} + \Delta\varepsilon_{\delta2} + \Delta\varepsilon_{\gamma1} + \Delta\varepsilon_{\gamma2} + \varepsilon(\infty)_{\text{relax}}.$$
(3)

To minimize the dependence on a model used for data fitting, herein, we focus only on the slow water relaxation (Fig. 1c; Nyquist plot for $\varepsilon^*_{\gamma1}(\omega)$), which accounts for ~80% of the total relaxation intensity (Supplementary Table 1):

$$\varepsilon^*_{\gamma1}(\omega) = \varepsilon_{\gamma1}(\infty) + \frac{\varepsilon_{\gamma1}(s) - \varepsilon_{\gamma1}(\infty)}{1+j\omega\tau_{\gamma1}}.$$
(4)

Because this relaxation component is sufficiently far in frequency from the other components, its high and low-frequency limits can be

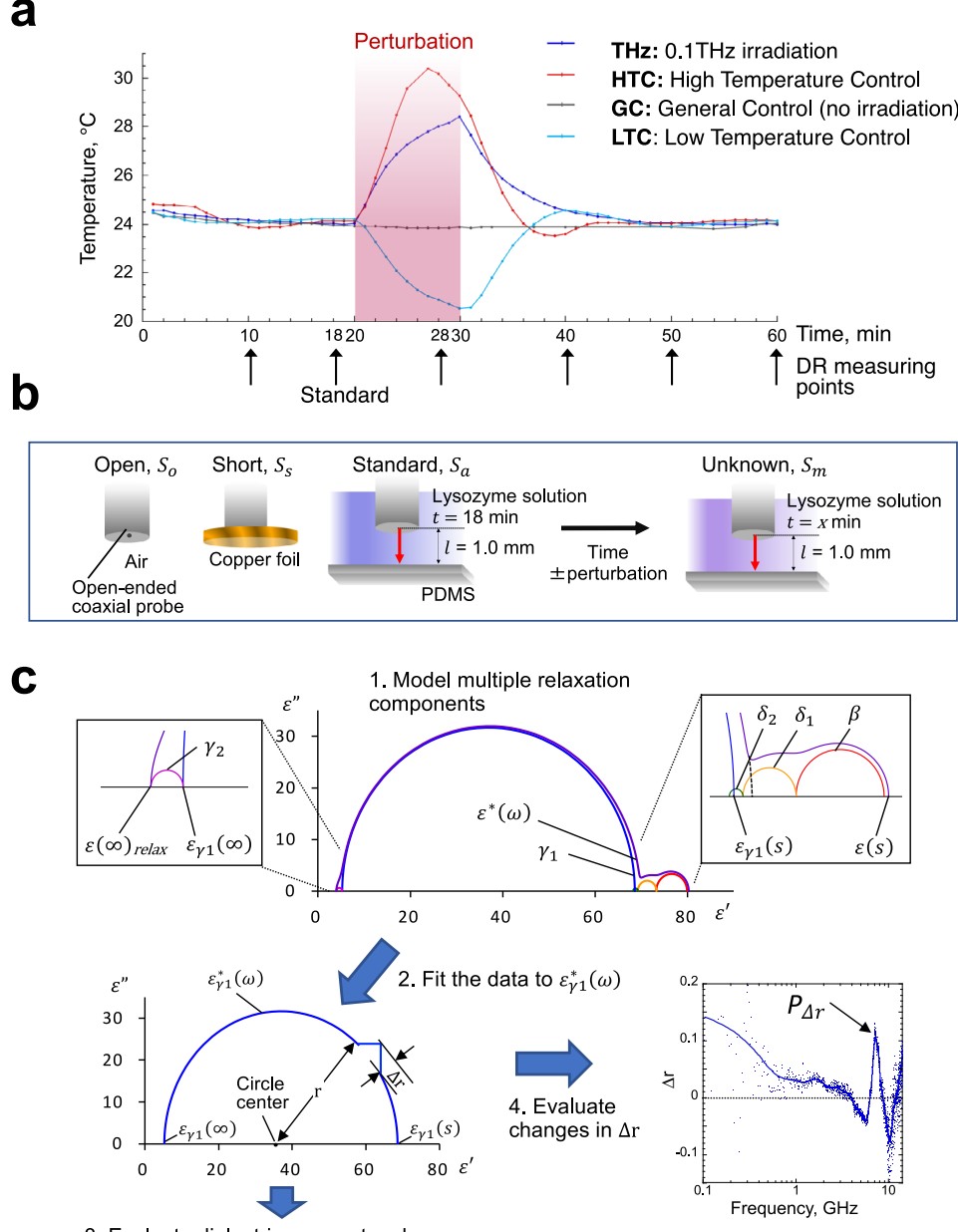

**Fig. 1 | Dielectric spectroscopic measurements and analysis. a** Time courses of the measurements of lysozyme solutions subjected to different perturbations. Temperatures measured in real time are represented on the vertical axis. The period of each perturbation caused by 0.1 THz irradiation (THz) or conduction heating/cooling (HTC/LTC) is indicated by red shading. **b** Method of time-lapse measurement. Initially, the complex dielectric permittivity of the Unknown sample was determined via Open, Short, and Standard calibration of the probe surface according to Eq. (7). **c** Relaxation analysis for the polydisperse liquid. The as-obtained spectra of the real and imaginary parts of complex permittivity were analyzed based on the Nyquist plot. In the multiple relaxation components consisting of lysozyme solution (top: $\beta$, $\delta_1$, $\delta_2$, $\gamma_1$ and $\gamma_2$), we analyzed the single Debye relaxation function of $\varepsilon^*_{\gamma1}(\omega)$ to calculate the dielectric parameters ($\varepsilon_{\gamma1}(s)$, $\varepsilon_{\gamma1}(\infty)$, and $f_{c\gamma1}$) and the shifts ($\Delta r$, deformed display) from the Debye relaxation model (bottom). The peak of $\Delta r$ is indicated by $P_{\Delta r}$.

obtained in the same manner as for $\varepsilon(s)$ and $\varepsilon(\infty)_{relax}$ by approximating the dielectric property within a semicircular complex plane with a radius $r = \{\varepsilon_{\gamma1}(s) - \varepsilon_{\gamma1}(\infty)\}/2$ (Fig. 1c):

$$\varepsilon_{\gamma1}(s) \approx \Delta\varepsilon_{\gamma1} + \Delta\varepsilon_{\gamma2} + \varepsilon(\infty)_{relax}, \qquad (5)$$

$$\varepsilon_{\gamma1}(\infty) \approx \Delta\varepsilon_{\gamma2} + \varepsilon(\infty)_{relax.} \qquad (6)$$

Of these approximate frequency limits, $\varepsilon_{\gamma1}(s)$ and its neighboring values in the complex plane are slightly displaced from the actual values by the close frequency proximity of $\delta$ relaxations (Fig. 1c, upper

right inset, blue and dashed lines). However, this does not affect the essential conclusions shown below, because we obtained the same results qualitatively using two different concentrations of lysozyme solutions, which had different relaxation strengths of $\beta$, $\delta_1$, and $\delta_2$, altering $\varepsilon_{\gamma1}(s)$ (Fig. 2, Supplementary Figs. 4a and 5a, Supplementary Tables 1 and 2).

## Nonthermal excitation effect during sub-THz irradiation

Using the measurements of the lysozyme solution and pure water during irradiation or heating/cooling, we calculated the extrapolated dielectric permittivity $\varepsilon_{\gamma1}(s)$, $\varepsilon_{\gamma1}(\infty)$, and relaxation frequency $f_{c\gamma1}$, where $f$ is the frequency of the external electric field. The results were

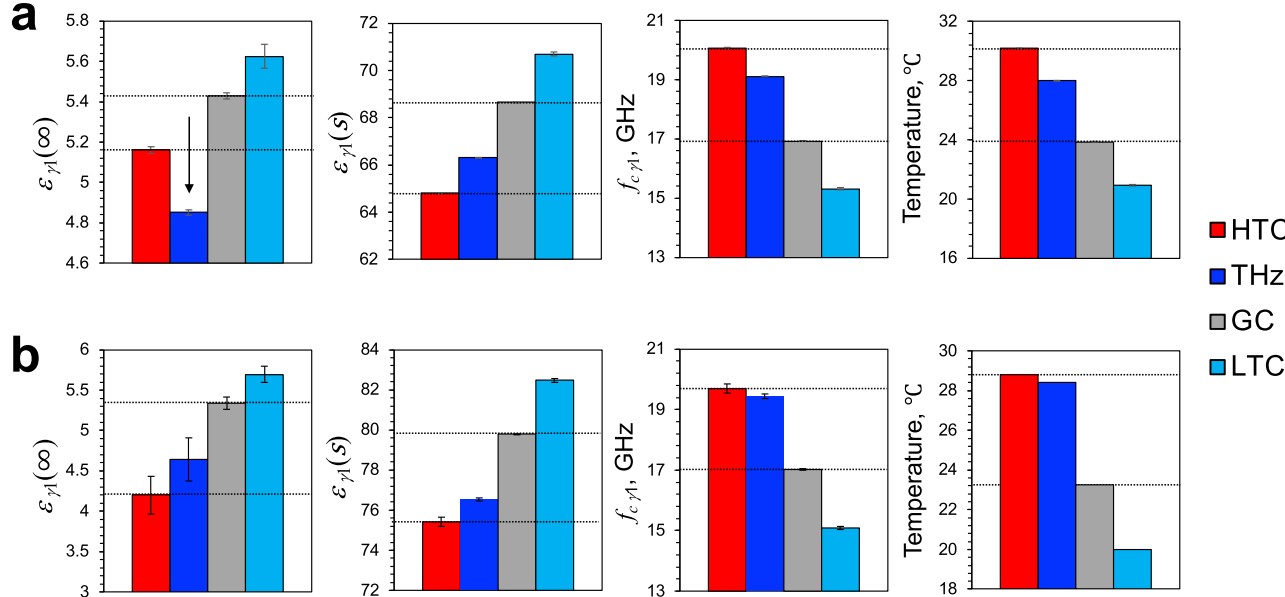

**Fig. 2 | Changes in dielectric parameters during 0.1 THz irradiation. a** The 9.1 wt% lysozyme solution. **b** Pure water. The means ± standard deviations of five measurements are shown. HTC and GC values are indicated by dashed lines. The 0.1-THz-induced decrease in $\varepsilon_{\gamma 1}(\infty)$ is indicated by an arrow. $\varepsilon_{\gamma 1}(\infty)$ represents the high-frequency limit of slow water relaxation, $\varepsilon_{\gamma 1}(s)$ represents the static slow water relaxation, and $f_{c\gamma 1}$ represents the slow water relaxation frequency.

plotted and compared with the profile of temperature changes (Fig. 2). If 0.1 THz irradiation is equivalent to heating, i.e., if isotropic thermal disturbance is dominantly detected during irradiation, any dielectric parameters of the irradiated sample (THz) should fall between those of HTC and GC. This case was applied in the profiles of $\varepsilon_{\gamma 1}(s)$ and $f_{c\gamma 1}$ in both the lysozyme and water samples (Fig. 2), indicating that the temperature is evenly influenced by irradiation. However, we observed a much larger decrease in $\varepsilon_{\gamma 1}(\infty)$ extrapolated to high frequencies for the lysozyme sample than would be predicted from the temperature increase (Fig. 2a). The amplitude of the decrease in $\varepsilon_{\gamma 1}(\infty)$ by irradiation became smaller when the lysozyme concentration was lowered from 9.1 wt% to 2.9 wt% and became larger when the measurement frequency was extended from 14 GHz to 40 GHz (Supplementary Fig. 4b). Therefore, the results obtained were lysozyme-dependent and are not artifacts of narrow bandwidth measurement. Notably, such a decrease in $\varepsilon_{\gamma 1}(\infty)$ by irradiation did not occur in water alone, and the presence of lysozyme significantly mitigated the decrease caused by temperature rise (Fig. 2), which implies that this observation is unrelated to the interference with the incident 0.1 THz field. In particular, the addition of lysozyme reduced the difference in $\varepsilon_{\gamma 1}(\infty)$ between HTC and LTC that was observed in water alone to ~30% (Fig. 2). These results indicate that the 0.1 THz radiation selectively perturbed the fast water dynamics that were generated by the interaction with lysozyme.

From Eq. (6), lowering $\varepsilon_{\gamma 1}(\infty)$ results in a decrease in either $\Delta\varepsilon_{\gamma 2}$ or $\varepsilon(\infty)_{relax}$ or both. A previous DR spectroscopic study that investigated the temperature dependence of the DR of water showed that $\Delta\varepsilon_{\gamma 2}$ decreases with increasing temperature by expanding the frequency range to 0.4 THz[38]. The result may depend on the measurement range and fitting model used due to its small abundance to bulk water $\Delta\varepsilon_{\gamma 1}$; however, a similar result has been obtained by using THz time-domain spectroscopy (THz-TDS) to analyze the relaxational and vibrational modes of water in the THz region[42]. These findings suggest that the origin of the temperature-dependent decrease in $\varepsilon_{\gamma 1}(\infty)$ observed in pure water can be approximated by $\Delta\varepsilon_{\gamma 2}$. By contrast, the $\Delta\varepsilon_{\gamma 2}$-like component observed as $\varepsilon_{\gamma 1}(\infty)$ in the lysozyme solution had a smaller temperature dependence than that of pure water (Fig. 2), and therefore, the relaxation origin may not be the same as that observed in pure water.

To evaluate $\Delta\varepsilon_{\gamma 2}$, we next used THz-TDS, which allows direct comparison of dielectric spectra between the lysozyme solution and pure water in the THz region (0.3–2.5 THz). As we have shown that the real part ($\varepsilon'$) of dielectric permittivity obtained using this method has a larger measurement error than the imaginary part ($\varepsilon''$)[39], we used only $\varepsilon''$ for the analysis (see Supplementary Fig. 6 for $\varepsilon'$). The lysozyme concentration was increased to 28.6 wt% for sensitively detecting any lysozyme-derived changes in relaxation modes. To evaluate $\varepsilon''$ only derived from relaxation modes of water that interact with lysozyme, the lysozyme-derived spectrum, which has been assigned as underdamped vibrational modes by Yamamoto et al.[15] (Supplementary Fig. 6a), was subtracted from the measured spectrum (Supplementary Fig. 6b), and it was further normalized by the fraction of water. Obtained $\varepsilon''$ spectra of water in the lysozyme solution were compared with that of pure water at different temperatures (Fig. 3a). These difference spectra revealed that the presence of lysozyme increased $\varepsilon''$ of the lysozyme-interacting water in the THz region that likely included the frequency of $\gamma_2$ relaxation (Fig. 3b), consistent with the prediction from the microwave DR measurement described above. Moreover, in the range of 20–35 °C, the $\varepsilon''$ overlapping with $\gamma_2$ peak (0.3–1 THz) tended to increase rather than decrease as the sample temperature increased (Fig. 3b), which was opposite to the temperature dependence of $\Delta\varepsilon_{\gamma 2}$ for pure water reported previously[38]. This result suggests that the origin of $\Delta\varepsilon_{\gamma 2}$ observed in the presence of lysozyme is different from that characterized in pure water, although the detected temperature dependence was at a level close to the measurement error (Fig. 3b).

Taken together, the combined microwave DR and THz-TDS measurements indicate that the incident 0.1 THz radiation selectively perturbed the water dynamics with increased mobility (i.e., fewer H-bonds) due to interaction with lysozyme. The result of the THz-TDS experiment also verified that our analysis method using the DR measurements below 14 GHz frequency, can approximately predict dielectric properties, including those in the THz region.

**Acceleration of protein hydration by sub-THz excitation**
In the DR measurement, we evaluated any shift from the single Debye-type relaxation due to 0.1 THz irradiation using $\Delta r$, which was defined as the difference between the radius ($r$) of the approximate

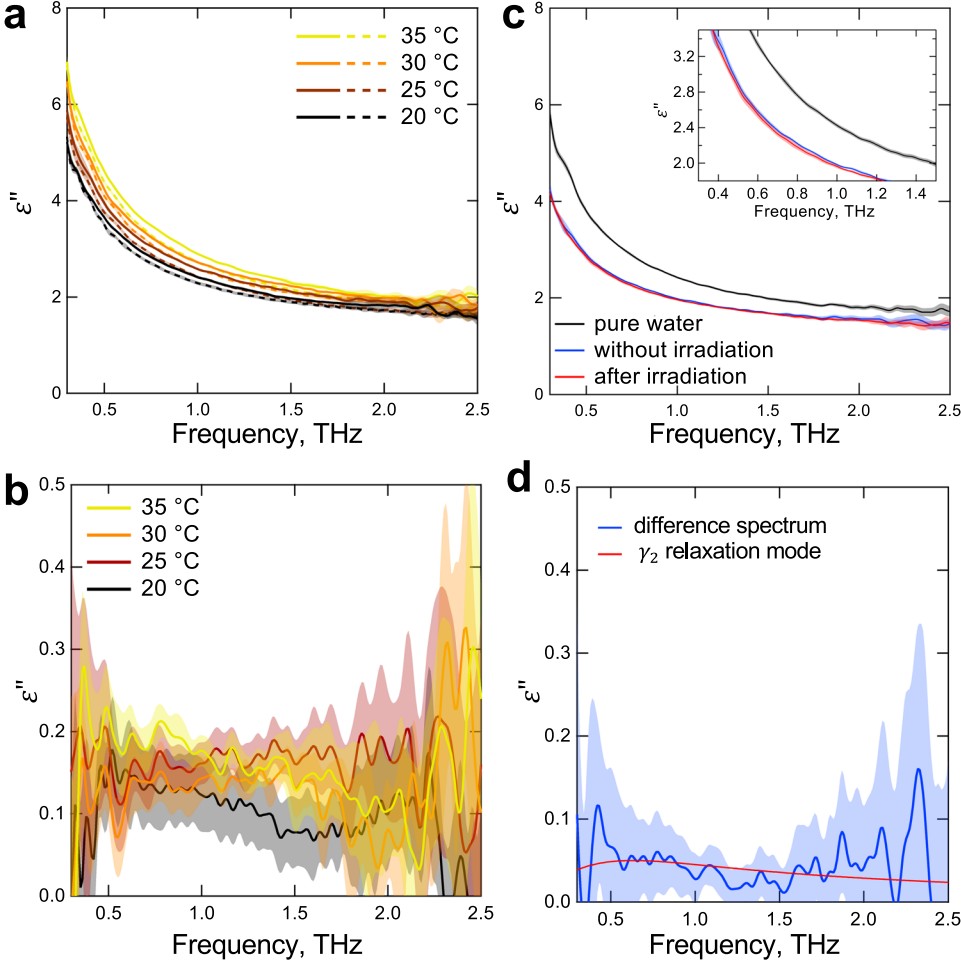

**Fig. 3 | Dielectric spectral analysis for the imaginary part of THz-TDS measurements. a** Solid lines represent spectra of water in the 28.6 wt% lysozyme solution at different temperatures. The spectra for dehydrated lysozyme[15] were subtracted from the raw spectra and were normalized by the fraction of water (=0.714). Dashed lines represent the corresponding spectra of pure water. **b** Subtracting the spectra of pure water (dashed lines of panel **a**) from those of water contained in the lysozyme solution (solid lines of panel **a**) gives spectra for lysozyme-interacting water. **c** Spectra for the lysozyme solution after 0.1 THz irradiation, the control sample without irradiation, and pure water at 25 °C. An enlarged view is shown in the inset. **d** Difference spectrum of the irradiated sample subtracted from non-irradiated control. The red line represents $\gamma_2$ (fast water) relaxation mode given by $\varepsilon''_{\gamma2}(\omega) = \Delta\varepsilon_{\gamma2}\omega\tau/(1 + j\omega^2\tau^2)$, where $\tau = 0.265$ ps, and $\Delta\varepsilon_{\gamma2}$ is arbitrary. All data are shown as means of four measurements. The measurement errors indicated by shading are given as follows. **a** $\sigma/0.714$ and $\sigma$ for water in the lysozyme solution and pure water, respectively, where $\sigma$ is the standard deviation of the four measurements. **c** b. **d** $\sqrt{\sigma_A^2 + \sigma_B^2}$, where $\sigma_A$ and $\sigma_B$ are standard deviations for each original spectrum before subtracting.

semicircle and the measured complex permittivity (Fig. 1c). Therefore, $\Delta r$ represents the deviation of the measured values from those calculated using the Debye model. Remarkably, upon plotting $\Delta r$ against frequency, a sharp signal appeared in the vicinity of 7–8 GHz, hereinafter referred to as $P_{\Delta r}$ (peak of the $\Delta r$ signal) (Fig. 1c, right graph). This signal was pronounced as the sample temperature increased at 28 min from the standard condition at 18 min (Fig. 4a), and thus, it would be attributable to a decrease in the dielectric permittivity of the sample. We intensively investigated the physical origin of the signal, which is described in Supplementary Notes and Supplementary Figs. 1–3. Briefly, due to the short $l$ in our measurement system, multiple reflections occurred between the sample cell and coaxial probe interfaces, which generated a standing wave of $\lambda/4$ through the sample (Supplementary Fig. 1c). As $l = 1.0$ mm, which is slightly longer than $\lambda/4$, a slight decrease in either $l$ or dielectric permittivity of the sample strongly finetuned the standing wave based on the relation, $\lambda \propto 1/\sqrt{\varepsilon}$. By finetuning the standing wave (i.e., generating the sharper and higher $P_{\Delta r}$), we could detect a decrease in the dielectric permittivity of the sample with high sensitivity.

If the irradiated lysozyme solution was far from thermal equilibrium and the history of irradiation accelerated to be or altered the equilibrium dielectric properties, the $P_{\Delta r}$ should vary upon irradiation. Therefore, we investigated the time-dependent changes in the $\Delta r$ profiles (Fig. 4a) and found that $P_{\Delta r}$ increased with time following 0.1 THz irradiation (Fig. 4a). In particular, we observed a qualitative difference between the irradiation and HTC with respect to the temporal variation of $P_{\Delta r}$: the height of $P_{\Delta r}$ for HTC reduced after heating, whereas for the irradiation, it rose even after the consequent temperature rise for both the low and high concentrations of lysozyme solutions (Fig. 4b). In addition, the position of $P_{\Delta r}$, which was observed at ~7 GHz during the temperature rise (28 min), shifted to a frequency ~1 GHz higher during the 40–60 min during which the temperature returned to room temperature (Fig. 4a, indicated by arrows). These results suggest that the dielectric permittivity of the lysozyme sample gradually decreased after irradiation for a reason(s) other than heating.

To explain $P_{\Delta r}$ in terms of DR phenomena of the lysozyme solution, we derived the difference between complex dielectric spectra before and after 0.1 THz irradiation or heating, from which the $\varepsilon'$ and $\varepsilon''$ parts of differential complex permittivity spectra were analyzed

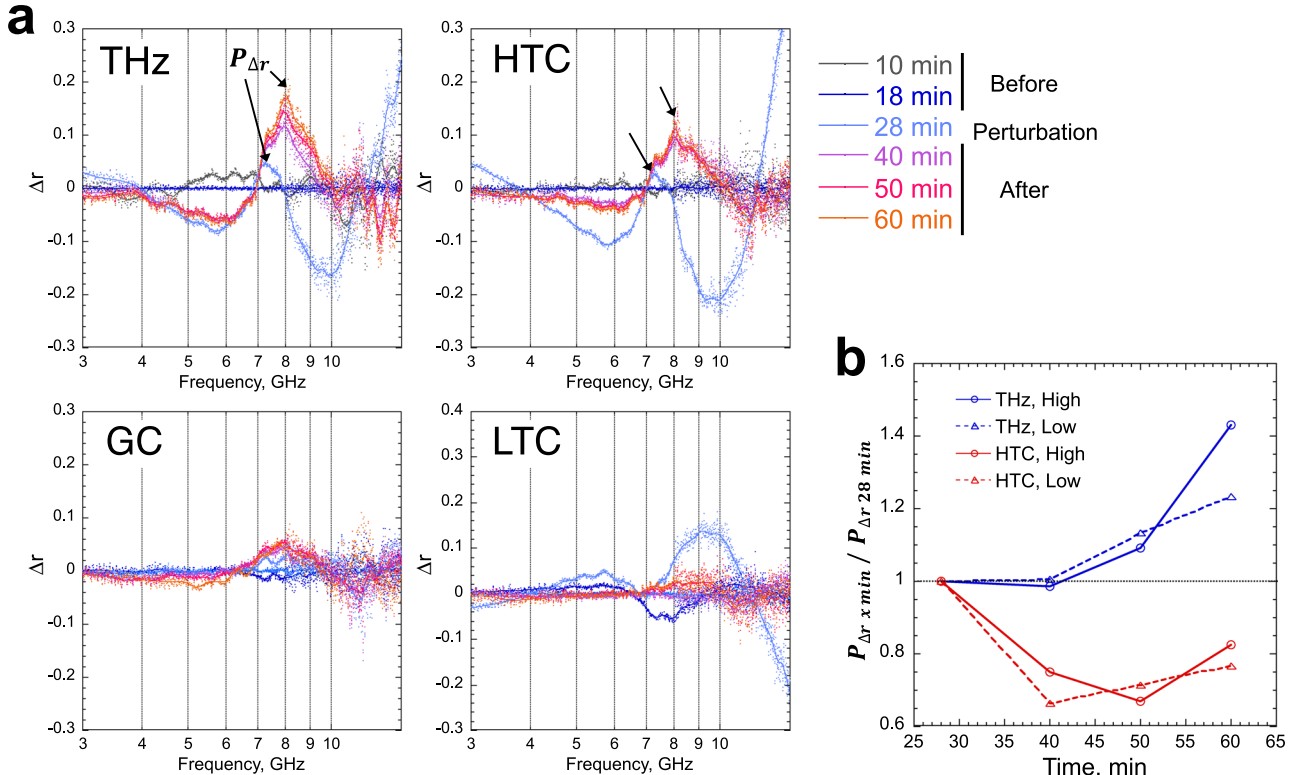

**Fig. 4 | Changes in peak of the Δr signal ($P_{\Delta r}$). a** Increase in $P_{\Delta r}$ (indicated by arrows) was accelerated following 0.1 THz irradiation (THz). The mean values of five measurements and moving average curves are shown. HTC, high-temperature control; GC, general control; LTC, low-temperature control. **b** Temporal change of the peak height of $P_{\Delta r}$ after irradiation or heating. $P_{\Delta r}$ is given by subtracting the baseline, which is defined as the mean of Δr (deviation of measured values from the Debye model) at each 1-GHz region on both sides of the peak. Results for high (9.1 wt%) and low (2.9 wt%) concentrations of lysozyme solutions are shown.

separately (Fig. 5a). The difference spectra reveal that irrespective of 0.1 THz irradiation, $P_{\Delta r}$ emerges in both $\varepsilon'$ (-7 GHz) and $\varepsilon''$ (-8 GHz) parts, as observed for the Δr spectra (Fig. 5a). Remarkably, the peak height of $P_{\Delta r}$ was correlated with a reduction in the $\varepsilon'$ part and an increase in the $\varepsilon''$ part in the frequency range of 0.3–3 GHz (Fig. 5b). In such a low-frequency region, $\varepsilon''$ of the differential spectrum increased beyond zero following 0.1 THz irradiation, whereas it was nearly zero following heating (Fig. 5a). This frequency range overlaps with $\delta$ relaxation of the hydration water identified in equilibrated aqueous lysozyme solutions[26,27]. Therefore, our observation supports the hypothesis that 0.1 THz irradiation would have increased the hydration water with reduced mobility (i.e., more H-bonds) in the non-equilibrated lysozyme solution, thereby increasing the abundance of $\delta$ relaxation ($\Delta\delta$) in the equilibrium or close to it. As the increased $\varepsilon''$ at the low frequency should result from a decreased $\varepsilon''$ at the higher frequencies according to Eq. (3), we employed THz-TDS to measure $\varepsilon''$ spectrum of the irradiated lysozyme solution in the THz region. We found a slight decrease in $\varepsilon''$ in the THz region, including the peak of $\gamma_2$ relaxation (Fig. 3c, d). This result is consistent with an interpretation that the strength of the fast water relaxation ($\Delta\varepsilon_{\gamma2}$) that was generated by the lysozyme may be shifted to the slow one ($\Delta\delta$) of the hydration water after irradiation.

**Atomic-level evaluation of the sub-THz excitation effects**
The above results suggest that 0.1 THz excitation could function analogously to shorten the time to reach hydration equilibrium after dissolving the dehydrated lysozyme powder in water. We thus compared the irradiation-dependent shift of $P_{\Delta r}$ between the lysozyme solution for 2 h and that dissolved in water for 24 h. As expected, the signal was much less sensitive to the irradiation for the samples dissolved for 24 h (Supplementary Figs. 5b, c).

In addition to microscopic changes in the hydrated water, structural changes in the protein such as reorganization of side chains or functional groups, or macroscopic changes in solution viscosity via protein aggregation or fibrillization, etc., may have occurred during the 24 h. The results of the DR and THz spectroscopies might depict not only the water molecules interacting with lysozyme, but also those perturbed by the counterion of lysozyme. Note that the lysozyme surface bears a positive charge (isoelectric point 11) because it was dissolved in pure water and the pH became ~3.4. Therefore, we evaluated all these possibilities at the atomic level via solution NMR spectroscopy using the 9.1 wt% lysozyme solution. In particular, we compared ¹H-¹³C heteronuclear single quantum coherence (HSQC) spectra of the lysozyme samples that experienced 0.1 THz irradiation (THz), with a temperature rise up to 31 °C (HTC) or room temperature (GC) during 20–30 min after water dissolution. In a ¹H-¹³C HSQC spectrum, correlation signals from aliphatic ¹H-¹³C pairs in lysozyme (CH, $CH_2$, and $CH_3$ groups) are observed at high resolution, whose chemical shifts and line shapes reflect site-specific structure and dynamics information. We particularly focused on $CH_3$ groups, as their high sensitivity owing to three equivalent protons with a preferable transverse relaxation property are beneficial to sensitive observation of natural abundance (-1%) ¹³C-¹H pairs.

Significant chemical shift differences were not observed among spectra of THz, HTC, and GC samples measured at both 3 h and 24 h (Supplementary Fig. 7a). Thus, the structural changes that strongly affect chemical shifts, including reorientation of sidechain dihedral angles and reorganization of aromatic rings, were not induced in lysozyme. This is not surprising, considering the case for the same irradiation to yeast ubiquitin[20]. Rather, it is more likely that the change in the protein–water interactions would have induced conformational

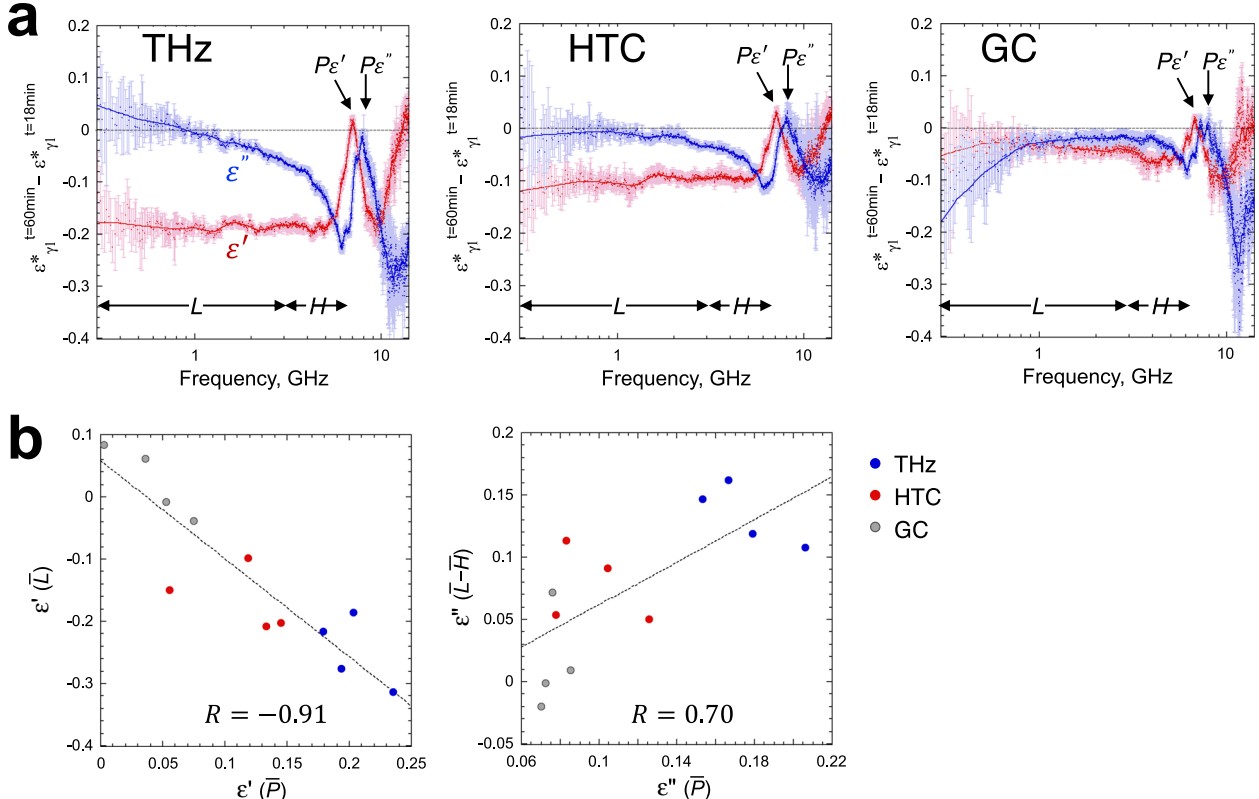

**Fig. 5 | Relation between peak of the $\Delta r$ signal ($P_{\Delta r}$) and complex permittivity ($\varepsilon^*$). a** Difference complex dielectric spectra before ($t = 18$ min) and after the perturbation ($t = 60$ min). The data for 2.9 wt% lysozyme sample are shown as a representative. *Low* (*L*: 0.3–3 GHz) and high (*H*: 3–6.5 GHz) frequency regions and peak heights ($P_{\varepsilon'}$ and $P_{\varepsilon''}$: -7 and -8 GHz for the real and imaginary parts, respectively) are indicated by arrows. The real and imaginary peak heights were obtained by subtracting the means of $\varepsilon'$ and $\varepsilon''$ in 4–5 GHz as the background, respectively. Data are shown with a moving average curve of the mean of five measurements ±

errors given by $\sqrt{\sigma_A^2 + \sigma_B^2}$ (For detail, see the legend of Fig. 3d). If there is no change in the spectra at the two time points, the spectra is on the dashed line. **b** Correlations of the frequency-averaged value of $P$ ($\bar{P}$) with those of *L* ($\bar{L}$, for real part, left) or $L–H$ ($\bar{L} - \bar{H}$, for imaginary part, right). The four plots for 0.1 THz irradiation (THz), heating (HTC), and control (GC) are derived from DR measurements performed with 2.9 wt% and 9.1 wt% lysozyme solutions, producing difference spectra of $\varepsilon^*_{t=60min} - \varepsilon^*_{t=18min}$ and $\varepsilon^*_{t=50min} - \varepsilon^*_{t=18min}$ at each concentration. The linear fitting of the data (dashed line) with a correlation coefficient $R$ is shown.

heterogeneity of sidechains, which was reflected in the NMR signal intensity due to an enhanced transverse relaxation rate.

Irrespective of irradiation, a slight uniform increase of methyl $^1$H-$^{13}$C signal intensity throughout the protein molecule was observed until ~3 h after dissolution (Supplementary Fig. 7b). This may be due to an enhanced rotational motion of the entire protein molecule, owing to gradual dissociation among proteins after being dissolved in water. Note that this observation is opposite to the cases of proteins developing aggregation or fibrillation. Such a gradual increase in the signal intensity hampers sensitive detection of any microscopic changes in the protein–water interactions. Therefore, we measured the spectra 3 h after water dissolution as the first timepoint and compared them to the measurements at 24 h.

To determine whether local conformational (or dynamical, the same applies hereinafter) changes of lysozyme caused by irradiation or heating proceed via similar or different pathways until 24 h, we performed a pairwise correlation analysis of methyl signals between various combinations (Fig. 6a and Supplementary Fig. 8). We plotted the positive and negative correlations above and below the GC measurements at 3 h, respectively, where the time (horizontal) axis was also added (Fig. 6a). We found a positive correlation between THz at 3 h and GC at 24 h, indicating a similar structural and hydration environment between these two conditions (Fig. 6a: A vs. B). This finding is also consistent with the results of DR and THz spectroscopies mentioned above. Positive-negative reversal of correlation observed between THz and HTC at 3 h also revealed that these pathways from GC at 3 h are opposite (Fig. 6a: B vs. C). In addition, the loss of

correlation observed between THz and HTC at 24 h showed that the conformational states of lysozyme after 24 h depend on these histories (Fig. 6a: BE vs. CD).

Next, we examined the spatial distribution of amino acid residues involved in such overall correlations. The respective residues for GC-24 h, THz-3 h, or HTC-3 h, whose signal intensities were significantly increased or decreased relative to those of GC-3 h, were mapped to the crystal structure of lysozyme (Fig. 6b). For GC-24 h and THz-3 h, the residues with increased or decreased signal intensity tended to localize around the hydrophobic cavity[43] (Fig. 6b, labeled in ochre). Moreover, at THz-3 h, the changes in the deeper sections of the hydrophobic cavity were more pronounced than at GC-24 h (Fig. 6b: L8$^{\delta2}$, L17$^{\delta2}$, I55$^{\gamma1}$).

Overall, the NMR results strongly suggest that the irradiation effect observed in the DR and THz spectroscopies is related to lysozyme-water interactions and not to significant changes in the lysozyme structure and counterion hydration. More details of the findings are discussed below.

## Discussion

In this study, we developed a time-lapse DR measurement and analysis method that facilitates the sensitive detection of the quantitative changes in relaxation of hydration water that were perturbed by the 0.1 THz EM field (Fig. 1a). Uniquely, this method made it possible to approximately detect the irradiation-dependent shifts in the fast relaxation strength $\Delta\varepsilon_{\gamma2}$ of water in the presence of lysozyme. In particular, any shift in $\Delta\varepsilon_{\gamma2}$ can be evaluated by shifting the semicircle,

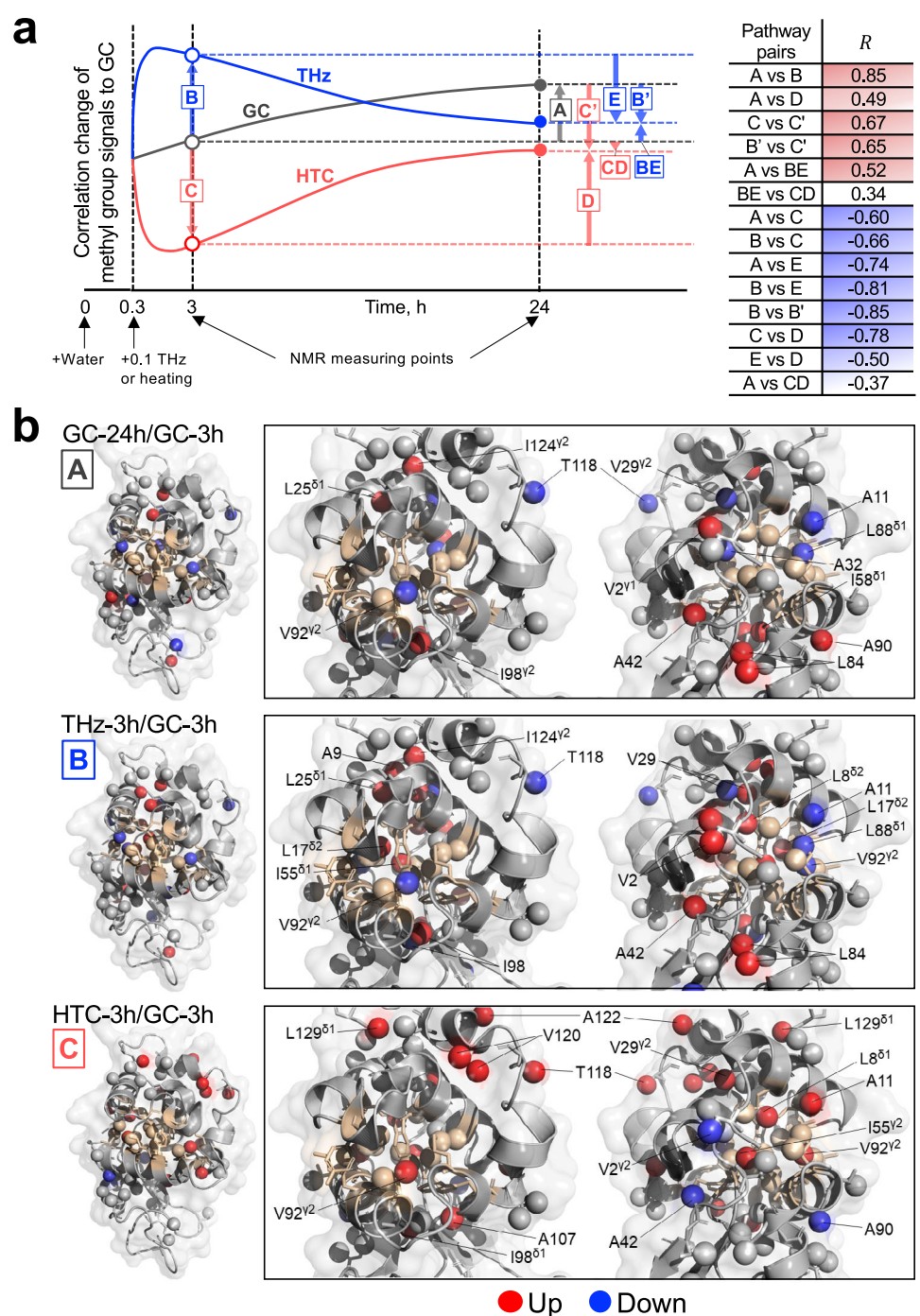

**Fig. 6 | NMR spectral analysis of lysozyme. a** Lysozyme proceeds to different pathways of conformational change after experiencing 0.1 THz irradiation (THz, blue) or temperature rise up to 31 °C (HTC, red). These pathways are shown schematically with respect to the GC pathway (gray). In the case of positive (or negative) correlation, the arrows are oriented in the same (or opposite) direction. The time-integrated ¹H-¹³C heteronuclear single quantum coherence spectra of lysozyme samples were obtained at 25 °C at 3–4 h (open circle) and 24–25 h (filled circle) after dissolution in water, from which methyl-group-derived signals were used for the analysis. A correlation coefficient *R* for any pair of two pathways is shown. **b** Structural Mapping. The tertiary lysozyme structure (left) as well as that focused on the hydrophobic cavity with 60° rotation (right) are shown (PDB accession code: 3EXD). The peptide backbone is also shown as a ribbon in a semi-transparent surface representation. The residue consisting of the hydrophobic cavity is colored ochre[43]. For the methyl group to be analyzed, the carbon atom is indicated by a sphere and amino acid residue is indicated by a stick. The residues were mapped in the lysozyme structure when $(I_{GC-24h}/I_{GC-3h}) - 1 > SD$ (red) or $1 - (I_{GC-24h}/I_{GC-3h}) > SD$ (blue), in the case of pathway A (top), where $I$ is the signal intensity (peak height) normalized to correct for the effect of differences in lysozyme concentration among samples and SD is the standard deviation of the ratio of analyzed methyl signals. The same applies to pathways B (middle) and C (bottom).

representing Debye-type relaxation of bulk water $\varepsilon^*_{\gamma1}(\omega)$, along the axis of the real part on the complex plane (Fig. 1c). In general, such an evaluation of $\Delta\varepsilon_{\gamma2}$ in pure water is difficult even in a more sophisticated analysis using broadband DR data because of its considerably small

strength (only ~2% of $\varepsilon(s)$) (Supplementary Table 1)[36–38,40–42]. The approximation used in this study also eliminates the arbitrary nature of fitting data to a function consisting of the sum of multiple Debye and vibration modes in typical DR analysis. Importantly, the irradiation-

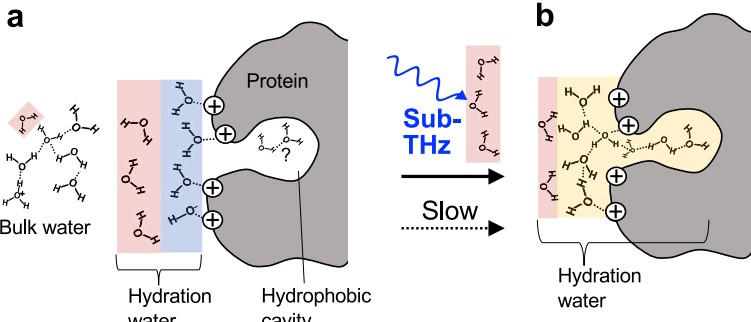

**Fig. 7 | Predicted sub-THz irradiation effect on hydration to heterogeneous protein surface.** Schematic illustration of hydration state 2 h after the lysozyme encounters liquid water (**a**), and that 24 h after the encounter or after sub-THz irradiation (**b**). Water molecules dominated by strong ion–dipole and ion–ion interactions (blue shading) generate H-bond-broken water molecules in the outer hydration layer (red shading) and prevent water entry into the hydrophobic cavity, where water molecules originally might have existed in isolation. Because of the acidic pH (-3.4), the positive charge of lysozyme can attract hydroxyl ions, and the bulk water contains sub-mM hydronium ions. The sub-THz excitation of the H-bond-broken (fast) water dynamics accelerates the reaction from state A to B to form hydrophobic hydration (ochre shading), leading to the increase in the number of H-bonds throughout the hydration layer.

dependent changes in $\Delta\varepsilon_{\gamma2}$ that were predicted from the DR analysis agree well with the direct measurements of the THz frequency range, including $\gamma_2$ relaxation by THz-TDS (Figs. 2 and 3).

Utilizing the resonant signal $P_{\Delta r}$ derived from a standing wave generated in the sample cell, we also successfully captured a small change (<0.1) in the dielectric permittivity remaining in the sample following the irradiation. In other words, our method introduces a type of Fabry–Pérot resonator to the coaxial probe reflection method in terms of evaluating the signal derived from multiple internal reflections of the incident EM wave (Supplementary Fig. 1). The result of $P_{\Delta r}$-based measurement suggests the presence of a slow chemical reaction that alters the hydration shell of lysozyme to reduce the dielectric permittivity (i.e., it reduces oriental polarization of water dipoles to the external field), which occurs over minutes to hours after dissolution of lysozyme powder in water, and 0.1 THz irradiation shortens the reaction (Figs. 4 and 5). To the best of our knowledge, such slow changes in the hydration structure have not been characterized rigorously and microscopically in proteins.

The irradiation history interpreted based on the dielectric permittivity of water was consistent with that interpreted from the change in methyl-group signal intensity on the lysozyme side measured via NMR. Although we could not determine how the rise and fall in the methyl-group signal intensity is related to its interaction with water molecules, we found that those changes upon irradiation are localized around the hydrophobic cavity of the protein (Fig. 6b). Combined with the result of DR measurements, such changes observed in the hydrophobic cavity could be interpreted as a progress in hydrophobic hydration, which normally takes more than 24 h from water dissolution of dehydrated lysozyme but was reached within 3 h upon irradiation.

Based on careful consideration of the results, the most plausible explanation for the sub-THz-irradiation effect on hydration at present is shown in Fig. 7. These views are inspired by the findings of hyper-mobile water detected in alkali halide, adenosine phosphate, and F-actin solutions via microwave DR spectroscopy[44–46], and they are further modified by a more recent finding via MD simulation combined with THz-TDS experiment, showing that $NH_4^+$ ions in the head group of a phospholipid break H-bonds between water molecules in the outer hydration layer, resulting in faster rotational relaxation[47,48]. Comparing the spectra of the lysozyme sample after irradiation with those of the non-irradiated sample using THz-TDS, the fast relaxation component ($\gamma_2$) derived from the hydration water appeared to be decreased for the irradiated sample (Fig. 3d). Furthermore, the microwave DR result suggests that the fast relaxation component that was reduced by irradiation was gradually replaced by an increase in the slow relaxation component ($\delta$) (Fig. 5). This replacement could be explained as

follows: Water molecules in the inner hydration layer are strongly attracted by the charged protein surface via ion–dipole interactions, while water molecules under dipole–dipole interactions in the outer hydration layer become relatively unstable by losing an H-bonding donor or acceptor. Such broken H-bonds in the outer hydration water have been detected as a blueshift of Raman spectra[49,50], which are also observed in our THz-TDS measurement of the lysozyme solution as the increased fast relaxation component (Fig. 3b). Because the lysozyme surface bears a positive charge at acidic pH, proton acceptors are eliminated for H-bonds of water molecules in the outer hydration layer. Therefore, the strong ion–dipole and additional (hydroxyl) ion–ion interactions at the surface can prevent the entry of any water molecules into the hydrophobic cavity by thermal motions (or the interactions of the water molecules originally isolated in the cavity with those outside, if any) (Fig. 7), as predicted by the NMR observation described above. Thus, the process to form hydrophobic hydration to increase the number of H-bonds in the overall hydration structure is expected to be largely delayed relative to forming the initial hydration structure dominated by ion–dipole interactions.

Our DR analysis suggests that the fast relaxation component of the hydration water is selectively perturbed by 0.1 THz irradiation in the lysozyme solution (Fig. 2a). This effect was not observed in water alone (Fig. 2b). Therefore, irradiation would excite and increase the amplitude of the fast dynamics generated by the protein–water interaction, which may enable the water molecules to overcome the strong ion–dipole interactions and to form a new H-bond network throughout the protein surface, including the hydrophobic cavity. Finally, for the above reasons, we infer that 0.1 THz irradiation can significantly shorten the process to form the hydrophobic hydration (Fig. 7). To obtain evidence for this inference, future study should directly observe detailed changes in the water H-bond network throughout the protein surface, including the hydrophobic cavity, before and after the irradiation.

We could not clarify the mechanism explaining how the sub-THz excitation energy remained localized in the fast hydration dynamics and the duration of this localization. However, it is possible that the excitation continues to occur during the pulse width (0.8 µs) of the incident 0.1 THz EM field, during which the energy barrier height or shape of the transition state of the hydration reaction may be altered. Another possibility may be related to the accumulated experimental evidence of long-lived excited low-frequency modes in protein structures, which were reported as a consequence of Fröhlich condensation[16,51–53]. Additionally, it was not obvious whether protein or hydrated water was the origin of the excited motion; however, these dynamics are collective and mutually influence each other as shown

previously[13–15]. It is clear from the NMR measurements that no significant lysozyme structural changes were observed following irradiation (Supplementary Fig. 7a). Future studies should focus on verifying whether 0.1 THz excitation can directly influence protein structures by developing a method such as NMR measurements during irradiation.

The phenomenon observed in this study appears to be specific to proteins having hydrophobic surfaces: We performed identical DR measurements using the following solutes (Supplementary Fig. 9): (i) cytochrome C, which is an amphiphilic globular protein having a hydrophobic core similar to that of lysozyme, (ii) salmon sperm DNA (500−1000 base pairs), which is also amphiphilic but has a highly hydrophilic surface owing to its sugar-phosphate backbone, and (iii) an ionic pair NaCl that provides a similar conductivity to the lysozyme solution. As expected, when cytochrome C was used as a solute for the 0.1 THz irradiation experiment, $P_{\Delta r}$ was observed to be as large as in the lysozyme solution. In contrast, using salmon sperm DNA and NaCl as the solute for the same experiment resulted in a weaker signal and no signal, respectively. This result is consistent with a notion that interactions between water dipoles and hydrophobic macromolecular surfaces may be the origin of $P_{\Delta r}$ induced by the 0.1 THz excitation. The dielectric permittivity of water has been reported to decrease near the hydrophobic surface[54,55], while the microscopic origin of the entropy loss upon hydrophobic hydration remains debatable[56].

Finally, from a biological point of view, our results using intense sub-THz EM field as the perturbation suggest that the heterogeneity in the hydration timescale that is induced by the protein surface may provide a clue to understand how the modulation of collective water dynamics on the surface contributes to achieving exquisite protein functions.

## Methods

### Materials
Crystalline powder of lysozyme from chicken egg white was purchased from Sigma-Aldrich Co. (Cat. No. 62971) and used without further purification. Ultrapure water (resistivity = 18.2 MΩ·cm at 25 °C, total organic carbon <5 ppb) used in all experiments was obtained from a Purelab Ultra Water Purification System (ELGA LabWater).

### Microwave DR spectroscopy
The dielectric properties of the aqueous solutions used in this study were measured at frequencies ranging from 100 MHz to 14 GHz and at ~32 μW by means of five measurement values using a VNA (Rhode & Schwarz, ZVB14). The measurement duration (i.e., time of applying the external EM field generated from VNA to the sample) per frequency point is estimated to be ~5 μs, according to the manufacturer's instructions. As the VNA-generated duration is similar to the pulse width of the incident 0.1 THz wave (~1 μs), if interference between the two EM fields occurs during 0.1 THz irradiation, there should be a large noise in the obtained dielectric spectrum at each measurement frequency point. However, no such noise was detected in our five measurements per sample (Supplementary Fig. 10). A SUS304 stainless steel coaxial line with $Z_0 = 50$ Ω and an outer diameter of 2.2 mm was used as an open-ended coaxial probe (PT292M2024486, KMCO). The VNA (Agilent, N5230A; 300 kHz−20 GHz) was also used for the same measurements with the same coaxial probe to verify that the measurements of interest were not dependent on the specific VNA and coaxial transmission line used. When the frequency range was expanded for the purpose of evaluating the resonant signal $P_{\Delta r}$ that was generated in the sample cell, another combination of the VNA (Anritsu, MS46131A; 20 MHz−43.5 GHz) and the coaxial probe (performance probe of N1501A dielectric probe kit, Keysight) was used for the measurements.

The coaxial probe was immersed into the 0.8 mL of liquid sample to be measured at the designated values of $l$ in a PDMS container

having an inner diameter of 20 mm, and the $S_{11}$ parameter was measured using the VNA. The complex permittivity ($\varepsilon_m^*$) of the Unknown sample was obtained by Open, Short, and Standard calibration of the probe surface (Fig. 1b), as follows.

$$\varepsilon_m^* = \frac{\varepsilon_a^*(S_m - S_o)(S_s - S_a) + (S_m - S_a)(S_o - S_s)}{(S_m - S_s)(S_o - S_a)}, \tag{7}$$

where the $S_{11}$ parameters that corresponded to the Open, Short, Standard (with known complex permittivity $\varepsilon_a^*$) and Unknown (with unknown complex permittivity $\varepsilon_m^*$) conditions were $S_o$, $S_s$, $S_a$, and $S_m$, respectively. The $\varepsilon_a^*(\omega)$ of the lysozyme solution was calculated with $\varepsilon_{\gamma1}(s) = 68.59$ (or 76.55), $\varepsilon_{\gamma1}(\infty) = 5.28$ (or 5.49), and $\tau = 5.88 \times 10^{-11}$ [s] as the parameters for the 9.1 wt% (or 2.9 wt%) solution (see Fig. 1c for details).

When the dielectric behavior is described by the single Debye-type relaxation function expressed in Eq. (4), complex permittivity $\varepsilon_{\gamma1}^*(\omega)$ follows a semicircular path in the Nyquist plane with angular frequency $\omega$ as the parameter, thereby transforming Eq. (4) into the circular form expressed as follows.

$$\left\{ \varepsilon_{\gamma1}'(\omega) - \frac{\varepsilon_{\gamma1}(s) - \varepsilon_{\gamma1}(\infty)}{2} \right\}^2 + \left\{ \varepsilon_{\gamma1}''(\omega) \right\}^2 = \left\{ \frac{\varepsilon_{\gamma1}(s) - \varepsilon_{\gamma1}(\infty)}{2} \right\}^2 \tag{8}$$

In Debye-type relaxation, $\varepsilon_{\gamma1}(s)$ and $\varepsilon_{\gamma1}(\infty)$ are located on the horizontal axis in the Nyquist plot. Therefore, these parameters were obtained by extrapolating the approximate semicircle to the horizontal axis based on the measurement between 100 MHz and 14 GHz. The relaxation frequency, $f_{c\gamma1}$, which is the vertex of the semicircle (the maximal point of $\varepsilon_{\gamma1}''$), was obtained using $\tan\theta$ of circumferential angle $\theta$.

### THz-TDS
The THz-TDS experiment and analysis were performed using a self-made instrument as described in ref. 39. Briefly, the attenuated total reflection (ATR) setup was applied for the sample cell. The temperature of the ATR sample cell, which contained 30 μl sample, was controlled within ±0.1 °C of the set temperature using a Peltier device.

### NMR experiments
Any aqueous samples of 9.1 wt% lysozyme were transferred into NMR microtubes with an outer diameter of 5 mm (Shigemi, Tokyo, Japan). Solvent water contained 10% D$_2$O for frequency lock. All NMR experiments were performed on a Bruker Avance III HD spectrometer equipped with a TXO triple-resonance cryoprobe (Bruker, Billerica, MA) at an $^1$H resonance frequency of 800 MHz at 298 K. For each sample, first, one-dimensional proton ($^1$H-1D) spectrum was acquired in ~1 min, with an acquisition time of 256 ms, a spectral width of 20 ppm, 16 scans, and an interscan delay of 1 s. Subsequently, a natural abundance $^1$H-$^{13}$C HSQC spectrum was obtained in 59 min, with acquisition times of 91.8 ms ($t2$, $^1$H) and 18.1 ms ($t1$, $^{13}$C), spectral widths of 14 and 26 ppm, respectively, 16 scans, and an interscan delay of 1 s. The carrier frequencies of $^1$H and $^{13}$C were 4.703 and 18 ppm, respectively. These experiments were performed in 3−4 h after dissolution. After the measurements, samples were incubated at room temperature for 20 h (i.e., 24 h total after dissolution) and the same sets of NMR experiments were performed. All the experiments were repeated two or more times, and representative results are shown in Fig. 6.

### NMR data processing and analysis
Time-domain data of the $^1$H-$^{13}$C HSQC experiments were multiplied by Gaussian and π/2-shifted squared sine window functions on $t2$ and $t1$ dimensions, respectively, followed by Fourier transformation and phase and baseline corrections in the TopSpin 3.4 software (Bruker).

Resultant frequency-domain ${}^1$H-${}^{13}$C HSQC spectra were analyzed using Sparky[57].

To account for the effects of subtle differences in lysozyme concentrations among samples, intensities of ${}^1$H-${}^{13}$C signals were normalized using the integral of the ${}^1$H-1D spectra, ranging from 12 to −3 ppm, within which amide, aromatic, and aliphatic proton resonances are included. These integrals are proportional to the lysozyme concentration. Note that the samples used here are aqueous solutions of lysozyme and do not contain any ingredients having ${}^1$H resonances and the region of the water resonance (4.75–4.65 ppm) was excluded from the integration of ${}^1$H-1D spectra.

Among the aliphatic ${}^1$H-${}^{13}$C correlations in 2D ${}^1$H-${}^{13}$C HSQC spectra, we analyzed those of methyl groups of Ala, Ile, Leu, Met, Thr, and Val (AILMTV) residues, which provide higher net magnetization and preferable transverse relaxation properties. Lysozyme has 61 AILMTV methyl groups, which distribute throughout the overall structure (see Fig. 6b), and thus it can be used for site-resolved interpretation of the structural changes. We transferred site-specific resonance assignments of ${}^1$H-${}^{13}$C resonances from the literature[58]. Among these, four resonances were excluded from further analyses due to signal overlapping. Thus, the remaining 57 resonances were served for further analyses.

### Sub-THz source and irradiation

The 0.1 THz radiation was directed toward the sample under the PDMS cell using a klystron-based sub-THz source[20]. A schematic of the experimental setup is illustrated in Supplementary Fig. 1a. The THz source comprised a W-band oscillator, preamplifier, isolator, direct reading attenuator (CAR-1050-01, WiseWave), klystron (Extended Interaction Klystron VKB2461, CPI Inc.), and pyramidal horn antenna. All devices were connected through rectangular waveguides, WR-10. This source can generate radiation in the frequency range of $95 \pm 0.25$ GHz, has a 10-kHz repetition rate, and generates a square wave of 0.8-μs pulse width. The emitted radiation is roughly collimated by a quartz plano-convex lens and monochromated by a bandpass filter with a center frequency of 95 GHz (MMBPF40, Joint Technology Development Platform Co., Ltd.). The radiation diameter was estimated to be approximately 20 mm at the sample position by measuring the full width at half maximum. Further details of the setup are described in ref. 20.

The sample solution was set on the top of total reflection mirror located at a distance of 200 mm from the bandpass filter (Supplementary Fig. 1a). The average power density of the wave that was radiated onto the sample was estimated using a calibrated thermal sensor (3A-P-THz, Ophir Optronics Solutions Ltd.).

The average electric field strength ($E\,[\mathrm{V/m}]$) and the energy density ($U\,[\mathrm{J/m^3}]$) at the irradiating surface of the aqueous sample ($\approx$ water) were estimated to be ~0.15 kV/m and 1.5 μJ/m³, respectively, according to the following relation.

$$E = \sqrt{\frac{P_s}{c\varepsilon_0\sqrt{\varepsilon'}}}, \qquad (9)$$

where $P_s\,[\mathrm{W/m^2}]$ is the measured surface power density ($=160\ \mathrm{W/m^2}$), $c\,[\mathrm{m/s}]$ is the speed of light in a vacuum, $\varepsilon_0\,[\mathrm{F/m}]$ is the electric constant, and $\varepsilon'\,(=7.7)$ is the real part of the dielectric permittivity at 0.1 THz frequency. $P_s$ and $U$ are related as follows.

$$U = \frac{P_s}{v}, \qquad (10)$$

where $v\,[\mathrm{m/s}]$ is the propagation velocity in water, given by $c/\sqrt{\varepsilon'}$. Although the field strength decays at a penetration depth of ~120 μm in the sample (absorption coefficient of the 0.1 THz EM field in water is 83 cm${}^{-1}$)[59], all molecules in the sample were repeatedly exposed to the field at the surface by diffusion during 10 min of irradiation.

## Data availability

All data generated or analyzed during this study are included in this published article (and its Supplementary information files).

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

## Acknowledgements

We thank Naoki Yamamoto and Keisuke Tominaga for kindly providing broadband dielectric spectra of dehydrated lysozyme and Katsuo Mogi for sharing the idea on using the PDMS container. We also thank Makoto Suzuki, Kimio Sumaru, Hiroshi Murakami, Kaito Sasaki, Shin Yagihara, and Hiroshi Ogawa for insightful discussions. We also thank Editage for English language editing. JSPS KAKENHI (project/area numbers JP22H04566 and JP20H03298 to M.I. and JP19H05717 to M.H.).

## Author contributions

Conceptualization: M.I., J.S., Y.T., and M.H. Methodology: J.S., M.I., Y.T., M.H., M.T., and K.T. Investigation: M.I., J.S., Y.T., M.H., M.T., and D.S. Resources: M.T., J.S., K.T., M.H., M.I., Y.T., and D.S. Visualization: M.I., J.S.,

Y.T., and M.H. Project administration: M.I., M.T., and K.T. Writing—original draft: M.I. and Y.T. (NMR part). Writing—review & editing: J.S., Y.T., M.H., M.T., D.S., and K.T.

## Competing interests

The authors declare no competing interests.
