## [Peer Review File · Nature Communications]

Nonthermal acceleration of protein hydration by sub-terahertz irradiationReviewers' comments:

Reviewer #1 (Remarks to the Author):

The manuscript titled "Nonthermal acceleration of protein hydration by sub-terahertz irradiation" involves the study of aqueous proteins that are investigated using dielectric spectroscopy, and specifically probes the influence of terahertz radiation and heating on the dielectric response of the samples. The authors claim that the dielectric results imply that 0.1 THz radiation is able to effectively alter the hydration dynamics surrounding the protein, in a sufficiently different manner than what occurs under heating. This builds on many years of studies involving terahertz dynamics of solvated water molecules.

I have no doubts regarding the experimental results, methodologies, or the quality of the data. However, my biggest issue with this manuscript is that it is not very clear that the results justify the conclusions, specifically regarding the amount of molecular-level details the authors extract from the experimental data, without much supporting evidence. For example, the authors attempt to conclude from broadband dielectric measurements precisely what the molecules are doing, how they are rearranging and relaxing, without much further evidence. Instead of a "slow hydration reaction occurring over hours to minutes" (which is very broadly defined by the authors), why can't it be one of the plethora of other flavors of slow relaxations that are occurring? How are the authors so sure that the origin of their response does not come from some protein dynamics, reorganization of side chains, reorganization of functional groups, etc etc etc? Overall, I find the atomic-level description lacking, and therefore in the end this work is mostly incremental and based on conjecture. A fine work, but some more evidence to the precise details would make this paper much more impactful. Therefore, as it stands, I cannot recommend publication in Nature Communications, and suggest a more specialized journal.

Reviewer #2 (Remarks to the Author):

This paper reports changes in the dielectric spectra of lysozyme solutions from with and without 0.1 THz-radiation. They observed some changes in the difference spectra of the lysozyme solutions and attributed this change to nonthermal acceleration of hydration. Since the discussion in the paper is very qualitative and lacks quantitative aspects, I suggest they include more quantitative discussion in the manuscript. For example, since they observed absolute values in the difference spectra (Figure 4), it becomes more convincing if they can estimate the absolute values by considering the radiation power of the 0.1 THz radiation, the interaction between water molecules and the radiation, the penetration depth of the radiation in the solution, so on. In page 8 the authors say that the dielectric behavior of the lysozyme solution follows the Debye relaxation. This is not true, as many authors reported, one of them is ref. 32, it is described in terms of multi-Debye function. The frequency of 14 GHz is not high enough in the dielectric measurements for aqueous solutions. What is the physical meaning of "epsilon(infinity)" obtained by extrapolation?

Reviewer #3 (Remarks to the Author):

The paper reports a novel interaction of 0.1 THz radiation with lysozyme solutions that can be considered as non-thermal in origin. In part, their research is motivated by a similar study that used NMR to probe the heterogeneous water dynamics occurring at the protein water interface for ubiquitin ¹. It also echoes predictions made by H. Fröhlich in the late 60s of enhanced excitations of bio macromolecules by sub-THz radiation ².

While the authors have taken into consideration most of the pitfalls of the experimental system they use, including the existence of standing waves from the base of the probe to the bottom of the sample

(reflected by the sample/PDMS interface), one is still left with a difficulty to really accept the results. The measurement is in the frequency band 11 MHz to 14 GHz. Basically, they are measuring the main bulk water response and whatever effect the solute has on the solvent. At the temperatures used (24 °C - 29 °C) the dielectric peak of water, even with its solute is around 19 GHz – 23 GHz ³. The solute will cause a red shift to the peak ⁴, but not a serious one. This means that the authors at best measured only a low frequency wing of the relaxation. However, they then extrapolate their results to the high frequencies to find ϵ_{∞} , where they claim that significant changes take place. To do so they employ a simple Debye function. This is methodologically difficult to accept. It is also well known that as solutes are added to water, the nature of the relaxation in this frequency range shift towards a Cole-Cole dependence ⁵, further loosening the validity of the extrapolation. The use of calibration of the lysozyme solution before and after irradiation, may indeed demonstrate an effect similar to what the authors proclaim, however without a strong justification for this extrapolation one cannot accept the conclusion.

References

1. Tokunaga, Y. et al. Nonthermal excitation effects mediated by sub-terahertz radiation on hydrogen exchange in ubiquitin. *Biophysical Journal* 120, 2386–2393 (2021).
2. Fröhlich, H. Long-range coherence and energy storage in biological systems. *International Journal of Quantum Chemistry* 2, 641–649 (1968).
3. Ellison, W. J. Permittivity of Pure Water, at Standard Atmospheric Pressure, over the Frequency Range 0–25 THz and the Temperature Range 0–100 °C. *Journal of Physical and Chemical Reference Data* 36, 1–18 (2007).
4. Cametti, C., Marchetti, S., Gambi, C. M. C. & Onori, G. Dielectric Relaxation Spectroscopy of Lysozyme Aqueous Solutions: Analysis of the δ -Dispersion and the Contribution of the Hydration Water. *J. Phys. Chem. B* 115, 7144–7153 (2011).
5. Levy, E., Puzenko, A., Kaatze, U., Ben Ishai, P. & Feldman, Y. Dielectric spectra broadening as the signature of dipole-matrix interaction. II. Water in ionic solutions. *The Journal of Chemical Physics* 136, 114503-114503-6 (2012).

Please note that the comments are in blue, and the responses are in black.

Reviewers' comments:

Reviewer #1 (Remarks to the Author):

The manuscript titled "Nonthermal acceleration of protein hydration by sub-terahertz irradiation" involves the study of aqueous proteins that are investigated using dielectric spectroscopy, and specifically probes the influence of terahertz radiation and heating on the dielectric response of the samples. The authors claim that the dielectric results imply that 0.1 THz radiation is able to effectively alter the hydration dynamics surrounding the protein, in a sufficiently different manner than what occurs under heating. This builds on many years of studies involving terahertz dynamics of solvated water molecules.

I have no doubts regarding the experimental results, methodologies, or the quality of the data. However, my biggest issue with this manuscript is that it is not very clear that the results justify the conclusions, specifically regarding the amount of molecular-level details the authors extract from the experimental data, without much supporting evidence. For example, the authors attempt to conclude from broadband dielectric measurements precisely what the molecules are doing, how they are rearranging and relaxing, without much further evidence. Instead of a "slow hydration reaction occurring over hours to minutes" (which is very broadly defined by the authors), why can't it be one of the plethora of other flavors of slow relaxations that are occurring? How are the authors so sure that the origin of their response does not come from some protein dynamics, reorganization of side chains, reorganization of functional groups, etc? Overall, I find the atomic-level description lacking, and therefore in the end this work is mostly incremental and based on conjecture. A fine work, but some more evidence to the precise details would make this paper much more impactful. Therefore, as it stands, I cannot recommend publication in Nature Communications, and suggest a more specialized journal.

First of all, we are grateful to Reviewer #1 for pointing the critical issues in our study. We agree with all the points made by Reviewer #1. In particular, dielectric permittivity, which we measured, is basically a macroscopic physical quantity, and therefore the atomic-level evaluation is necessary for precisely describing the sub-THz irradiation effects on hydration of lysozyme. We need to list and verify all possible interpretations of the results. In the revised manuscript, we performed comprehensive analysis combining microwave dielectric relaxation (DR) (Figs. 1, 2, 4 and 5) and THz time-domain spectroscopy (THz-TDS) (Fig. 3) and NMR spectroscopy (Fig. 6), which cross-validated the consistency of the results obtained by each spectroscopic method and any possible interpretations in the 0.1 THz irradiation effects. Please note that the THz-TDS and NMR are the experiments that are newly added in the revised manuscript.

We agree that in addition to microscopic changes in the hydrated water, structural changes in the protein such as reorganization of side chains or functional groups, or macroscopic changes in solution viscosity via protein aggregation or fibrillation, etc., may have occurred during hours (in the time course that we set in the measurements). Also, we later noticed that the results of dielectric spectroscopy might depict not only the water molecules interacting with lysozyme, but also those perturbed by the counterion of lysozyme. Therefore, we evaluated all these possibilities at the atomic level via solution NMR spectroscopy.

For more details on the results, please see the newly added Result subsection “*Atomic-level evaluation of the sub-THz excitation effects*” in the revised manuscript (lines 322-399, Fig. 6 and Supplementary Figs. S7 and S8).

The results of our NMR analysis are briefly described below. Overall, 0.1-THz irradiation history interpreted based on the dielectric permittivity of water (shown in our initially submitted manuscript) was consistent with that interpreted from the change in methyl-group signal intensity on the lysozyme side measured via NMR.

In detail, any significant chemical shift differences were not observed in the spectra between 0.1-THz irradiation and heating/ room temperature control samples measured both at 3 h and 24 h (**Fig. S7A**). This suggests that the conformational state of lysozyme is nearly identical throughout the experiments. Therefore, we investigated the changes in signal intensity. Signal intensity will decrease if the conformational heterogeneity is enhanced, because of a larger transverse relaxation rate. We performed a correlation analysis of methyl signals between various combinations of two pathways from the condition of room temperature to that of the 0.1-THz irradiation or heating (**Fig. 6A and Fig. S8**). We found a positive correlation between irradiation at 3 h and room temperature at 24 h, indicating that the structural environments of lysozyme, including hydrated water, are similar between them (Fig. 6A: A vs. B). This finding is also consistent with the results of the DR spectroscopies. Positive-negative reversal of correlation observed between irradiation and heating at 3 h also revealed that these pathways from room temperature at 3 h are opposite (Fig. 6A: B vs. C). We also found that the methyl-group signal intensity changes upon irradiation are localized around the hydrophobic cavity of the protein (Figure 6B).

The NMR results strongly suggest that the irradiation effect we observed in the DR and THz spectroscopies is related to lysozyme-water interactions but not to significant changes in lysozyme structure and the counterion hydration.

Fig. 6. NMR spectral analysis of lysozyme. (A) Lysozyme proceeds to different pathways of conformational change after experiencing 0.1-THz irradiation (THz, blue) or temperature rise up to 31°C (HTC, red). These pathways are shown schematically with respect to the GC pathway (gray). In the case of positive (or negative) correlation, the arrows are oriented in the same (or opposite) direction. The time-integrated ^1H - ^{13}C HSQC spectra of lysozyme samples were obtained at 25 °C at 3–4 h (open circle) and 24–

25 h (filled circle) after dissolution in water, from which methyl-group-derived signals were used for the analysis. A correlation coefficient R for any pair of two pathways is shown. **(B)** Structural Mapping. The tertiary lysozyme structure (left) as well as that focused on the hydrophobic cavity with 60° rotation (right) are shown (PDB accession code: 3EXD). The peptide backbone is also shown as a ribbon in a semi-transparent surface representation. The residue consisting of the hydrophobic cavity is colored ochre⁴³. For the methyl group to be analyzed, carbon atom is indicated by a sphere, and amino acid residue is indicated by a stick. The residues were mapped in the lysozyme structure when $(I_{GC-24h}/I_{GC-3h}) - 1 > SD$ (red) or $1 - (I_{GC-24h}/I_{GC-3h}) > SD$ (blue), in the case of pathway A (top), where I is the signal intensity (peak height) normalized to correct for the effect of differences in lysozyme concentration among samples. SD is the standard deviation of the ratio of analyzed methyl signals. The same applies to the pathways B (middle) and C (bottom).

Fig. S7. 2D ^1H - ^{13}C HSQC spectra (methyl region) of 9.1 wt% lysozyme. (A) Spectrum of the GC-3 h sample is shown upper left with assignments transferred from a reported data (Biological Magnetic Resonance Bank entry 4562). Spectra of the other conditions, GC-24 h (bottom left), THz-3 h (upper middle), THz-24 h (bottom middle), HTC-3 h (upper right), and HTC-24 h (bottom right) are superimposed in red on that of GC-3 h (black). It should be noted that the black contours of the GC-3 h are mostly masked below red contours of the superimposed spectra. (B) Averaged ratio of methyl peak heights for each of the spectra shown in the panel A over those in the spectrum obtained at 40 min after dissolution. The error bars indicate the standard deviations of ratio of all signals. *P*-values ($*P < 0.05 \times 10^{-3}$, $**P < 0.005$, $***P < 1 \times 10^{-5}$) of two-tailed t-test are shown for pairs with statistically significant differences to the the spectrum obtained at 40 min after dissolution.

Fig. S8. Correlation analysis for any pair of two pathways shown in Fig. 6A. For each methyl signal, the ratio of signal intensity at the end point to that at the starting point of each pathway is plotted. The error bar of a ratio of i -th methyl site for a pathway X-Y is derived from the signal-to-noise ratio (SNR) as follows:

$$Error_{i,X-Y} = (1/SNR_{i,X} + 1/SNR_{i,Y})/\sqrt{2}.$$

To make the definition of hydration that we used here more specific, we developed a new approach for the DR analysis that models multiple relaxation components of the aqueous lysozyme solution (lines 137-165, Fig. 1C and Eq. 1-6. Also, please see below our response to Reviewers #2 and #3). This analysis suggests that the fast relaxation component (i.e., hydration water with fewer H-bonds than bulk water) is decreased upon irradiation (**Figs. 2 and 3**). Notably, the fast relaxation component that was reduced by irradiation was gradually replaced by an increase in the slow relaxation component (i.e., more H-bonds than bulk water) (**Figs 3-5**). Based on the newly added data, this replacement could be explained as follows (**Fig.7**): Water molecules in the inner hydration layer are strongly attracted by the charged protein surface via ion-dipole interactions, while water molecules under dipole-dipole interactions in the outer hydration layer become relatively unstable by losing H-bonding donor or acceptor. Such broken H-bonds in the outer hydration water has been detected as a blueshift of Raman spectra, which are also observed in our THz-TDS measurement (**Fig. 3B**).

Fig. 7. Predicted sub-THz irradiation effect on hydration to heterogeneous protein surface. Schematic illustration of hydration state 2 h after the lysozyme encounters liquid water (**A**), and that 24 h after the encounter or after sub-THz irradiation (**B**). Water molecules dominated by strong ion–dipole and ion–ion–interactions (blue shading) generate H-bond-broken water molecules in the outer hydration layer (red shading) and prevent water entry into the hydrophobic cavity. Because of the acidic pH (~3.4), positive charge of lysozyme can attract the hydroxyl ion and the bulk water contains sub mM hydronium ion. The sub-THz excitation of the H-bond-broken (fast) water dynamics accelerates the reaction from state A to B to form hydrophobic hydration (ochre shading), leading to the increase in the number of H-bonds throughout the hydration layer.

For more details, please see Results and Discussion sections of the revised manuscript, where all changes from the previous manuscript are indicated in blue. We extensively revised the previous manuscript and almost all parts are rewritten.

Reviewer #2 (Remarks to the Author):

This paper reports changes in the dielectric spectra of lysozyme solutions from with and without 0.1 THz-radiation. They observed some changes in the difference spectra of the lysozyme solutions and attributed this change to nonthermal acceleration of hydration.

Since the discussion in the paper is very qualitative and lacks quantitative aspects, I suggest them include more quantitative discussion in the manuscript.

First of all, we are grateful to Reviewer #2 for suggesting essential issues that improve our study. We extensively revised the previous manuscript and almost all parts are rewritten. Please see the revised manuscript, where changes from the previous manuscript are indicated in blue.

We agree with the suggestion to include more quantitative discussion in the manuscript. We thus made two major revisions: (i) We constructed a new approach for the dielectric relaxation (DR) analysis that models multiple relaxation components of the aqueous lysozyme solution, allowing more accurate and quantitative evaluation of the changes in relaxation modes of hydration water with fewer H-bonds or more H-bonds compared to bulk water (lines 137-165, Fig. 1 and Eq. 1-6). The revision that we have made is related to a specific comment below provided by this reviewer, so we describe our detailed response to the specific comment. (ii) In the revised manuscript, we performed comprehensive analysis combining microwave DR (Figs. 1, 2, 4 and 5) and THz time-domain spectroscopy (THz-TDS) (Fig. 3) and NMR spectroscopy (Fig. 6), which cross-validated the consistency of the results obtained by each spectroscopic method and any possible interpretations in the 0.1 THz irradiation effects. THz-TDS and NMR are the experiments that are newly added in the revised manuscript. In particular, in the revised paper, the change in the fast relaxation component (i.e., hydration water with fewer H-bonds than bulk water) can be directly evaluated by THz-TDS that can provide the dielectric response in the range of 0.3-2.5 THz, allowing for a more quantitative discussion in the fast relaxation mode. Please see our major change in Discussion for detail (lines 402-493). We also quantify the correlation between the decrease in dielectric permittivity and the increase in $P\Delta r$ signal due to standing waves (Fig. 5B).

Please note that this study presents a finding of new phenomena regarding the interaction of the intense sub-THz electromagnetic field with water and protein molecules, which are largely unknown. Thus, what is most important is description of the phenomena as objective as possible (not relying on one likely model), based on the reproducibility and determining the error range of the measurement, and the cross-validation of the consistency of the observations by using multiple measurement methods with different principles. Therefore, we focus on these issues in this early stage of the findings of new phenomena. For this reason, it is not always appropriate to fit the data to a simple known model for quantification. If appropriate, we describe results qualitatively for

which quantification involves arbitrariness in model selection. (e.g., calculation of the change in the number of hydration water molecules from slight spectral changes shown in Fig. 3).

For example, since they observed absolute values in the difference spectra (Figure 4), it becomes more convincing if they can estimate the absolute values by considering the radiation power of the 0.1 THz radiation, the interaction between water molecules and the radiation, the penetration depth of the radiation in the solution, so on.

We thank this suggestion. We agree that in order to evaluate the 0.1-THz irradiation effect based on the orientation polarization of water and lysozyme molecules, one would need information on the electric field strength and energy density at the irradiating surface and its penetration depth in the sample. Therefore, we added the following in Materials and Methods (lines 623-634):

The average electric field strength (E [V/m]) and the energy density (U [J/m³]) at the irradiating surface of the aqueous sample (\approx water) was estimated to be ~ 0.15 kV/m and 1.5 μ J/m³, respectively, according to the following relation.

$$E = \sqrt{\frac{P_s}{c\epsilon_0\sqrt{\epsilon'}}} \quad (8)$$

where the P_s [W/m²] is the measured surface power density ($= 160$ W/m²), the c [m/s] is the speed of light in vacuum, the ϵ_0 [F/m] is the electric constant and the ϵ' ($= 7.7$) is the real part of the dielectric permittivity at 0.1-THz frequency. The P_s and the U are related as follows.

$$U = \frac{P_s}{v}, \quad (9)$$

where v [m/s] is the propagation velocity in water given as $c/\sqrt{\epsilon'}$. Although the field strength decays at a penetration depth of ~ 120 μ m in the sample (absorption coefficient of the 0.1 THz EM field in water is 83 cm⁻¹)⁵⁹, all molecules in the sample will be repeatedly exposed to the field at the surface by diffusion during 10 min of irradiation.

Please note that detail of the relationship between the quantity of the electric field strength and sub-THz excitation effects on protein hydration are not discussed in this paper. Assuming some suitable model and performing simulations would be a subject for future work.

In page 8 the authors say that the dielectric behavior of the lysozyme solution follows the Debye relaxation. This is not true, as many authors reported, one of them is ref. 32, it is described in terms of multi-Debye function.

We agree with this point. In the revised manuscript, we constructed a new approach to analyze the dielectric behavior of the lysozyme solution by the multi-Debye function as follows (lines 137-

165):

The dielectric response of aqueous lysozyme solution in MHz–GHz frequency regions includes protein-derived relaxation at ~10 MHz (β) and two hydration water-derived relaxations at ~0.1 and ~4 GHz (δ_1 and δ_2), respectively, at room temperature^{26,27}. In addition, slow and fast relaxations from bulk water appear at ~20 GHz and ~150~600 GHz (γ_1 and γ_2 , the exact frequency is debatable), respectively³⁶⁻⁴¹. Therefore, the distribution of multiple relaxations involving Debye processes can be expressed as Eq. 1 (**Figure 1C**; Nyquist plot for $\varepsilon^*(\omega)$).

$$\varepsilon^*(\omega) = \varepsilon(\infty)_{relax} + \frac{\Delta\varepsilon_\beta}{1+j\omega\tau_\beta} + \frac{\Delta\varepsilon_{\delta_1}}{1+j\omega\tau_{\delta_1}} + \frac{\Delta\varepsilon_{\delta_2}}{1+j\omega\tau_{\delta_2}} + \frac{\Delta\varepsilon_{\gamma_1}}{1+j\omega\tau_{\gamma_1}} + \frac{\Delta\varepsilon_{\gamma_2}}{1+j\omega\tau_{\gamma_2}}, \quad (1)$$

where $\Delta\varepsilon$ is the strength of each relaxation, and $\varepsilon(\infty)_{relax}$ is the apparent high-frequency limit in the Debye-type relaxation comprising all vibrational components ($\Delta\varepsilon_{vib}$) of the higher frequency regions and the high-frequency limit:

$$\varepsilon(\infty)_{relax} = \sum \Delta\varepsilon_{vib} + \varepsilon(\infty). \quad (2)$$

Taking the limit of $\omega \rightarrow 0$, Eq. 3 gives static dielectric constant:

$$\varepsilon(s) = \Delta\varepsilon_\beta + \Delta\varepsilon_{\delta_1} + \Delta\varepsilon_{\delta_2} + \Delta\varepsilon_{\gamma_1} + \Delta\varepsilon_{\gamma_2} + \varepsilon(\infty)_{relax}. \quad (3)$$

To minimize the dependence on a model used for data fitting, herein, we focus only on the slow water relaxation (**Figure 1C**; Nyquist plot for $\varepsilon_{\gamma_1}^*(\omega)$), which accounts for ~80% of the total relaxation intensity (**Table S1**):

$$\varepsilon_{\gamma_1}^*(\omega) = \varepsilon_{\gamma_1}(\infty) + \frac{\varepsilon_{\gamma_1}(s) - \varepsilon_{\gamma_1}(\infty)}{1+j\omega\tau_{\gamma_1}}. \quad (4)$$

Because this relaxation component is sufficiently far in frequency from the other components, its high and low-frequency limits can be obtained in the same manner as for $\varepsilon(s)$ and $\varepsilon(\infty)_{relax}$ by approximating the dielectric property within a semicircular complex plane with a radius $r = \{\varepsilon_{\gamma_1}(s) - \varepsilon_{\gamma_1}(\infty)\}/2$ (**Figure 1C**):

$$\varepsilon_{\gamma_1}(s) \approx \Delta\varepsilon_{\gamma_1} + \Delta\varepsilon_{\gamma_2} + \varepsilon(\infty)_{relax}, \quad (5)$$

$$\varepsilon_{\gamma_1}(\infty) \approx \Delta\varepsilon_{\gamma_2} + \varepsilon(\infty)_{relax}. \quad (6)$$

Of these approximate frequency limits, $\varepsilon_{\gamma_1}(s)$ and its neighboring values in the complex plane are slightly displaced from the actual values by the close frequency proximity of δ relaxations (**Figure 1C**, upper right inset, blue and dashed lines). However, this does not affect the essential conclusions shown below, because we obtained the same results qualitatively using two different concentrations of lysozyme solutions, which have different relaxation strengths of β , δ_1 , and δ_2 , altering $\varepsilon_{\gamma_1}(s)$ (**Figure 2**, **Figure S4A**, **Tables S1 and S2**).

Fig. 1 (C) Relaxation analysis for the polydisperse liquid. The as-obtained spectra of the real and imaginary parts of complex permittivity were analyzed based on the Nyquist plot. In the multiple relaxation components consisting of lysozyme solution (top: β , δ_1 , δ_2 , γ_1 and γ_2), we analyzed the single Debye relaxation function of $\varepsilon_{\gamma_1}^*(\omega)$ to calculate the dielectric parameters ($\varepsilon_{\gamma_1}(s)$, $\varepsilon_{\gamma_1}(\infty)$, and fc_{γ_1}) and the shifts (Δr , deformed display) from the Debye relaxation model (bottom). The peak of Δr is indicated by $P_{\Delta r}$.

The frequency of 14 GHz is not high enough in the dielectric measurements for aqueous solutions.

We thank this comment, which is important to explain the validity of the new DR analysis that we have developed. First, an expansion of the measurement frequency from 14 GHz to 40 GHz allowed us to obtain the similar and more pronounced 0.1-THz irradiation effect (**Figure S4**). Second, the prediction of the changes in the dielectric response for the high frequency region ($\gg 14$ GHz) obtained with the DR measurements up to 14 GHz have been validated with THz-TDS, which can measure the dielectric response in the THz region (0.3-2.5 THz), with consistent conclusions. Thus, although our measurement range was narrow, the results predicted from the DR analysis agreed well with the direct measurements of the THz frequency range by THz-TDS (as discussed in lines 402-415). For more information, please see below descriptions in the revised manuscript (lines 167-246).

Using the measurements of the lysozyme solution and pure water during irradiation or heating/cooling, we calculated the extrapolated dielectric permittivity $\varepsilon_{\gamma_1}(s)$, $\varepsilon_{\gamma_1}(\infty)$, and relaxation frequency fc_{γ_1} , where f is the frequency of the external electric field. The results were plotted and compared with the profile of temperature changes (**Figure 2**). If 0.1 THz irradiation is equivalent to heating, i.e., if isotropic thermal disturbance is dominantly detected during irradiation, any dielectric parameters of the irradiated sample (THz) should fall between those of HTC and GC. This case was applied in the profiles of $\varepsilon_{\gamma_1}(s)$ and fc_{γ_1} in both the

lysozyme and water samples (**Figure 2**), indicating that the temperature is evenly influenced by irradiation. However, we observed a much larger decrease in $\varepsilon_{\gamma_1}(\infty)$ extrapolated to high frequencies for the lysozyme sample than would be predicted from the temperature increase (**Figure 2A**). The amplitude of the decrease in $\varepsilon_{\gamma_1}(\infty)$ by irradiation became smaller when the lysozyme concentration was lowered from 9.1 wt% to 2.9 wt% and became larger when the measurement frequency was extended from 14 GHz to 40 GHz (**Figure S4**). Therefore, the results obtained are lysozyme-dependent and are not artifacts of narrow bandwidth measurement. Notably, such a decrease in $\varepsilon_{\gamma_1}(\infty)$ by irradiation did not occur in water alone, and the presence of lysozyme significantly mitigated the decrease by temperature rise (**Figure 2**), also implying that this observation is unrelated to the interference with the incident 0.1 THz field. In particular, the addition of lysozyme reduced the difference in $\varepsilon_{\gamma_1}(\infty)$ between HTC and LTC that was observed in water alone to $\sim 30\%$ (**Figure 2**). These results indicate that the 0.1-THz radiation selectively perturbed the fast water dynamics that were generated by the interaction with lysozyme.

Fig. 2. Changes in dielectric parameters of the lysozyme solution (A) and pure water (B) during 0.1-THz irradiation. The mean values of five measurements \pm standard deviations are shown. HTC and GC values are indicated by dashed lines. The 0.1-THz-induced decrease in $\varepsilon_{\gamma_1}(\infty)$ is indicated by an arrow.

From Eq. 6, lowering of $\varepsilon_{\gamma_1}(\infty)$ results in a decrease in either $\Delta\varepsilon_{\gamma_2}$ or $\varepsilon(\infty)_{relax}$ or both. A previous DR spectroscopic study that investigated the temperature dependence of the dielectric relaxation of water showed that $\Delta\varepsilon_{\gamma_2}$ decreases with increasing temperature by expanding the frequency range to 0.4 THz^{38,42}. The result may depend on the measurement range and fitting model used due to its small abundance to bulk water $\Delta\varepsilon_{\gamma_1}$; however, a similar result has been obtained by using THz time-domain spectroscopy (THz-TDS) to analyze the relaxational and vibrational modes of water in the THz region⁴². These findings suggest that the origin of the

temperature-dependent decrease in $\varepsilon_{\gamma_1}(\infty)$ observed in pure water can be approximated by $\Delta\varepsilon_{\gamma_2}$. By contrast, the $\Delta\varepsilon_{\gamma_2}$ observed in the lysozyme solution had a smaller temperature dependence than that of pure water (**Figure 2**), and therefore, the relaxation origin may not be the same as that observed in pure water.

To evaluate $\Delta\varepsilon_{\gamma_2}$ we next used THz-TDS, which allows direct comparison of dielectric spectra between the lysozyme solution and pure water in the THz region (0.3–2.5 THz). As we have shown that the real part (ε') of dielectric permittivity obtained using this method has a larger measurement error than the imaginary part (ε'')³⁹, we used only ε'' for the analysis (see Figure S6 for ε'). The lysozyme concentration was increased to 28.6 wt% for sensitively detecting any lysozyme-derived changes in relaxation modes. To evaluate ε'' only derived from relaxation modes of water that interact with lysozyme, the lysozyme-derived spectrum, which has been assigned as underdamped vibrational modes by Yamamoto et al.¹⁵ (Figure S6A), was subtracted from the measured spectrum (Figure S6B), and it was further normalized by the fraction of water. Obtained ε'' spectra of water in the lysozyme solution were compared with that of pure water at different temperatures (**Figure 3A**). These difference spectra revealed that the presence of lysozyme increased ε'' of the lysozyme-interacting water in the THz region that likely included the frequency of γ_2 relaxation (**Figure 3B**), consistent with the prediction from the microwave DR measurement described above. Moreover, in the range of 20–35 °C, the ε'' overlapping with γ_2 peak (0.3–1 THz) tended to increase rather than decrease as the sample temperature increased (**Figure 3**), which was opposite to the temperature dependence of $\Delta\varepsilon_{\gamma_2}$ for pure water reported previously³⁸. This result suggests that the origin of $\Delta\varepsilon_{\gamma_2}$ observed in the presence of lysozyme is different from that characterized in pure water, although the detected temperature-dependence was at a level close to the measurement error (**Figure 3**).

Taken together, the combined microwave DR and THz-TDS measurements indicate that the incident 0.1-THz radiation selectively perturbed water dynamics with increased mobility (i.e., fewer H-bonds) due to interaction with lysozyme. The result of THz-TDS experiment also verified that our analysis method using the DR measurements below 14 GHz frequency, can approximately predict dielectric properties, including those in the THz region.

Fig. 3. Dielectric spectral analysis for the imaginary part of THz-TDS measurements. (A) Solid lines represent spectra of water in the 28.6 wt% lysozyme solution at 20 °C (purple), 25 °C (blue), 30 °C (green), and 35 °C (red). The spectra for dehydrated lysozyme¹⁵ were subtracted from the raw spectra and were normalized by the fraction of water (= 0.714). Dashed lines represent the corresponding spectra of pure water. (B) Subtracting the spectra of pure water (dashed lines of panel A) from those of water contained in the lysozyme solution (solid lines of panel A) gives spectra for lysozyme-interacting water. Color code is the same as shown in panel A. (C) Spectra for the lysozyme solution after 0.1-THz irradiation (red), the control sample without irradiation (blue), and pure water (black) at 25 °C. An enlarged view is shown in the inset. (D) Difference spectrum of the irradiated sample (red) subtracted from non-irradiated control (blue). A red line represents γ_2 relaxation mode given by $\varepsilon''_{\gamma_2}(\omega) = \Delta\varepsilon_{\gamma_2}\omega\tau/(1+j\omega^2\tau^2)$, where $\tau = 0.265$ ps and $\Delta\varepsilon_{\gamma_2}$ is arbitrary. All data are shown as means of four measurements. The measurement errors indicated by shading are given as follows. Panel A: $\sigma/0.714$ and σ for water in the lysozyme solution and pure water, respectively, where σ is the standard deviation of the four measurements. Panel C: σ . Panel B and D: $\sqrt{\sigma_A^2 + \sigma_B^2}$, where σ_A and σ_B are standard deviations for each original spectrum before subtracting.

What is the physical meaning of “epsilon(infinity)” obtained by extrapolation?

As described above, our analysis combining microwave DRS and THz-TDS suggests that the change in the “epsilon(infinity)” ($= \epsilon_{\gamma 1}(\infty)$ for the revised manuscript) approximately represents the change in the relaxation strength of the fast water dynamics ($\Delta\epsilon_{\gamma 2}$) that were generated by the interaction with lysozyme.

Reviewer #3 (Remarks to the Author):

The paper reports a novel interaction of 0.1 THz radiation with lysozyme solutions that can be considered as non-thermal in origin. In part, their research is motivated by a similar study that used NMR to probe the heterogeneous water dynamics occurring at the protein water interface for ubiquitin 1. It also echoes predictions made by H. Fröhlich in the late 60s of enhanced excitations of bio macromolecules by sub-THz radiation 2.

While the authors have taken into consideration most of the pitfalls of the experimental system they use, including the existence of standing waves from the base of the probe to the bottom of the sample (reflected by the sample/PDMS interface), one is still left with a difficulty to really accept the results.

First of all, we are grateful to Reviewer #3 for carefully reading our manuscript and providing us insightful comments. We agree with the point that one is still left with a difficulty to really accept the results, and thus we extensively revised the initially submitted manuscript and almost all parts are rewritten. Please see the revised manuscript, where changes from the previous manuscript are indicated in blue.

We agree with the point that in the initially submitted manuscript, it was necessary to more adequately validate the methodology that we have developed. In the revised manuscript, we performed comprehensive analysis combining microwave dielectric relaxation (DR) (Figs. 1, 2, 4 and 5) and THz time-domain spectroscopy (THz-TDS) (Fig. 3) and NMR spectroscopy (Fig. 6), which cross-validated the consistency of the results obtained by each spectroscopic method and any possible interpretations in the 0.1 THz irradiation effects. Please note that the THz-TDS and NMR are the experiments that are newly added in the revised manuscript. We also clarified the correlation between the decrease in dielectric permittivity and the increase in PAr signal due to the standing waves that were generated in the sample cell (Fig. 5B).

These new results and analyses that we made in the revised manuscript are clearly written in the revised Results section and Supplementary text, which are also summarized in Discussion as follows (lines 402-435):

In this study, we developed a time-lapse DR measurement and analysis method that facilitates the sensitive detection of the quantitative changes in relaxation of hydration water that were perturbed by the 0.1-THz EM field (**Figure 1A**). Uniquely, this method made it possible to approximately detect the irradiation-dependent shifts in the fast relaxation strength $\Delta\varepsilon_{\gamma_2}$ of water in the presence of a lysozyme. In particular, any shift in $\Delta\varepsilon_{\gamma_2}$ can be evaluated by parallelly shifting the semicircle on the complex plane, representing Debye-type relaxation of bulk water $\varepsilon_{\gamma_1}^*(\omega)$, along the axis of real part (**Figure 1C**). In general, such an evaluation of $\Delta\varepsilon_{\gamma_2}$ in pure water is difficult even in a more sophisticated analysis using broadband DR data because of its considerably small strength (only $\sim 2\%$ of $\varepsilon(s)$) (**Table S1**)^{36-38,40-42}. The approximation used in this study would also eliminate the arbitrary nature of fitting data to a function consisting of the sum of multiple Debye and vibration modes in typical DR analysis. Importantly, the irradiation-dependent changes in $\Delta\varepsilon_{\gamma_2}$ that were predicted from the DR analysis agree well with the direct measurements of the THz frequency range, including γ_2 relaxation by THz-TDS (**Figures 2 and 3**).

Utilizing the resonant signal $P_{\Delta r}$ derived from a standing wave that is generated in the sample cell, we also successfully captured a small change (<0.1) in the dielectric permittivity remaining in the sample following the irradiation. In other words, our method introduces a type of Fabry–Pérot resonator to the coaxial probe reflection method in terms of evaluating the signal derived from multiple internal reflections of the incident EM wave (**Figure S1**). The result of $P_{\Delta r}$ -based measurement suggests the presence of a slow chemical reaction altering hydration shell of lysozyme to reduce the dielectric permittivity (i.e., reduce oriental polarization of water dipoles to the external field), which occurs over minutes to hours after dissolution of lysozyme powder in water, and 0.1-THz irradiation shortens the reaction (**Figures 4 and 5**). To the best of our knowledge, such slow changes in the hydration structure have not been characterized rigorously and microscopically in proteins.

The irradiation history interpreted based on the dielectric permittivity of water was consistent with that interpreted from the change in methyl-group signal intensity on the lysozyme side measured via NMR. Although we could not determine how the rise and fall in the methyl-group signal intensity is related to its interaction with water molecules, we found that those changes upon irradiation are localized around the hydrophobic cavity of the protein (**Figure 6B**). Combined with the result of DR measurements, such changes observed in the hydrophobic cavity could be interpreted as a progress in hydrophobic hydration, which normally takes more than 24 h to reach from water dissolution of dehydrated lysozyme but reached within 3 h upon irradiation.

The measurement is in the frequency band 11 MHz to 14 GHz. Basically, they are measuring the main bulk water response and whatever effect the solute has on the solvent. At the temperatures used (24 °C - 29 °C) the dielectric peak of water, even with its solute is around 19 GHz – 23 GHz. The solute will cause a red shift to the peak, but not a serious one. This means that the authors

at best measured only a low frequency wing of the relaxation.

We understand the concern raised by Reviewer #3. Please see above our response to Reviewer #2, where we provided our detailed response to this issue. Briefly, first, expansion of the measurement frequency from 14 GHz to 40 GHz allowed us to obtain the similar and more pronounced 0.1-THz irradiation effect (**Figure S4**). Second, the extrapolated prediction of the high frequency region (>14 GHz) obtained with DR measurements up to 14 GHz have been validated with THz-TDS, which can measure the dielectric response in the THz region (0.3-2.5 THz), with consistent conclusions. Thus, although our measurement range was narrow, the result extrapolatively predicted from the DR analysis agreed well with the direct measurements of the THz frequency range by THz-TDS. For more information, please see Results section of the revised manuscript (lines 167-246).

However, they then extrapolate their results to the high frequencies to find ϵ_∞ , where they claim that significant changes take place. To do so they employ a simple Debye function. This is methodologically difficult to accept.

We agree with this comment. In the revised manuscript, we constructed a new approach for the DR analysis that models multiple relaxation components of the aqueous lysozyme solution as follows (lines 137-165), and we obtained clearer sub-THz irradiation effect than using the previous simple one.

The dielectric response of aqueous lysozyme solution in MHz–GHz frequency regions includes protein-derived relaxation at ~10 MHz (β) and two hydration water-derived relaxations at ~0.1 and ~4 GHz (δ_1 and δ_2), respectively, at room temperature^{26,27}. In addition, slow and fast relaxations from bulk water appear at ~20 GHz and ~150~600 GHz (γ_1 and γ_2 , the exact frequency is debatable), respectively³⁶⁻⁴¹. Therefore, the distribution of multiple relaxations involving Debye processes can be expressed as Eq. 1 (**Figure 1C**; Nyquist plot for $\epsilon^*(\omega)$).

$$\epsilon^*(\omega) = \epsilon(\infty)_{relax} + \frac{\Delta\epsilon_\beta}{1+j\omega\tau_\beta} + \frac{\Delta\epsilon_{\delta_1}}{1+j\omega\tau_{\delta_1}} + \frac{\Delta\epsilon_{\delta_2}}{1+j\omega\tau_{\delta_2}} + \frac{\Delta\epsilon_{\gamma_1}}{1+j\omega\tau_{\gamma_1}} + \frac{\Delta\epsilon_{\gamma_2}}{1+j\omega\tau_{\gamma_2}}, \quad (1)$$

where $\Delta\epsilon$ is the strength of each relaxation, and $\epsilon(\infty)_{relax}$ is the apparent high-frequency limit in the Debye-type relaxation comprising all vibrational components ($\Delta\epsilon_{vib}$) of the higher frequency regions and the high-frequency limit:

$$\epsilon(\infty)_{relax} = \sum \Delta\epsilon_{vib} + \epsilon(\infty). \quad (2)$$

Taking the limit of $\omega \rightarrow 0$, Eq. 3 gives static dielectric constant:

$$\epsilon(s) = \Delta\epsilon_\beta + \Delta\epsilon_{\delta_1} + \Delta\epsilon_{\delta_2} + \Delta\epsilon_{\gamma_1} + \Delta\epsilon_{\gamma_2} + \epsilon(\infty)_{relax}. \quad (3)$$

To minimize the dependence on a model used for data fitting, herein, we focus only on the slow water relaxation (**Figure 1C**; Nyquist plot for $\varepsilon_{\gamma_1}^*(\omega)$), which accounts for $\sim 80\%$ of the total relaxation intensity (**Table S1**):

$$\varepsilon_{\gamma_1}^*(\omega) = \varepsilon_{\gamma_1}(\infty) + \frac{\varepsilon_{\gamma_1}(s) - \varepsilon_{\gamma_1}(\infty)}{1 + j\omega\tau_{\gamma_1}}. \quad (4)$$

Because this relaxation component is sufficiently far in frequency from the other components, its high and low-frequency limits can be obtained in the same manner as for $\varepsilon(s)$ and $\varepsilon(\infty)_{relax}$ by approximating the dielectric property within a semicircular complex plane with a radius $r = \{\varepsilon_{\gamma_1}(s) - \varepsilon_{\gamma_1}(\infty)\}/2$ (**Figure 1C**):

$$\varepsilon_{\gamma_1}(s) \approx \Delta\varepsilon_{\gamma_1} + \Delta\varepsilon_{\gamma_2} + \varepsilon(\infty)_{relax}, \quad (5)$$

$$\varepsilon_{\gamma_1}(\infty) \approx \Delta\varepsilon_{\gamma_2} + \varepsilon(\infty)_{relax}. \quad (6)$$

Of these approximate frequency limits, $\varepsilon_{\gamma_1}(s)$ and its neighboring values in the complex plane are slightly displaced from the actual values by the close frequency proximity of δ relaxations (**Figure 1C**, upper right inset, blue and dashed lines). However, this does not affect the essential conclusions shown below, because we obtained the same results qualitatively using two different concentrations of lysozyme solutions, which have different relaxation strengths of β , δ_1 , and δ_2 , altering $\varepsilon_{\gamma_1}(s)$ (**Figure 2, Figure S4A, Tables S1 and S2**).

Fig. 1 (C) Relaxation analysis for the polydisperse liquid. The as-obtained spectra of the real and imaginary parts of complex permittivity were analyzed based on the Nyquist plot. In the multiple relaxation components consisting of lysozyme solution (top: β , δ_1 , δ_2 , γ_1 and γ_2), we analyzed the single Debye relaxation function of $\varepsilon_{\gamma_1}^*(\omega)$ to calculate the dielectric parameters ($\varepsilon_{\gamma_1}(s)$, $\varepsilon_{\gamma_1}(\infty)$, and $f_{C_{\gamma_1}}$) and the shifts (Δr , deformed display) from the Debye relaxation model (bottom). The peak of Δr is indicated by $P_{\Delta r}$.

It is also well known that as solutes are added to water, the nature of the relaxation in this frequency range shift towards a Cole-Cole dependence ⁵, further loosening the validity of the extrapolation.

We understand this concern. In the revised manuscript, in addition to extending the measurement frequency from 14 GHz to 40 GHz, we validated the extrapolation-based prediction with the direct measurement of the THz frequency range by THz-TDS (**Fig. 3**). For more information, please see below descriptions in the revised manuscript (lines 167-246).

Using the measurements of the lysozyme solution and pure water during irradiation or heating/cooling, we calculated the extrapolated dielectric permittivity $\varepsilon_{\gamma 1}(s)$, $\varepsilon_{\gamma 1}(\infty)$, and relaxation frequency $fc_{\gamma 1}$, where f is the frequency of the external electric field. The results were plotted and compared with the profile of temperature changes (**Figure 2**). If 0.1 THz irradiation is equivalent to heating, i.e., if isotropic thermal disturbance is dominantly detected during irradiation, any dielectric parameters of the irradiated sample (THz) should fall between those of HTC and GC. This case was applied in the profiles of $\varepsilon_{\gamma 1}(s)$ and $fc_{\gamma 1}$ in both the lysozyme and water samples (**Figure 2**), indicating that the temperature is evenly influenced by irradiation. However, we observed a much larger decrease in $\varepsilon_{\gamma 1}(\infty)$ extrapolated to high frequencies for the lysozyme sample than would be predicted from the temperature increase (**Figure 2A**). The amplitude of the decrease in $\varepsilon_{\gamma 1}(\infty)$ by irradiation became smaller when the lysozyme concentration was lowered from 9.1 wt% to 2.9 wt% and became larger when the measurement frequency was extended from 14 GHz to 40 GHz (**Figure S4**). Therefore, the results obtained are lysozyme-dependent and are not artifacts of narrow bandwidth measurement. Notably, such a decrease in $\varepsilon_{\gamma 1}(\infty)$ by irradiation did not occur in water alone, and the presence of lysozyme significantly mitigated the decrease by temperature rise (**Figure 2**), also implying that this observation is unrelated to the interference with the incident 0.1 THz field. In particular, the addition of lysozyme reduced the difference in $\varepsilon_{\gamma 1}(\infty)$ between HTC and LTC that was observed in water alone to ~30% (**Figure 2**). These results indicate that the 0.1-THz radiation selectively perturbed the fast water dynamics that were generated by the interaction with lysozyme.

Fig. 2. Changes in dielectric parameters of the lysozyme solution (A) and pure water (B) during 0.1-THz irradiation. The mean values of five measurements \pm standard deviations are shown. HTC and GC values are indicated by dashed lines. The 0.1-THz-induced decrease in $\varepsilon_{\gamma_1}(\infty)$ is indicated by an arrow.

From Eq. 6, lowering of $\varepsilon_{\gamma_1}(\infty)$ results in a decrease in either $\Delta\varepsilon_{\gamma_2}$ or $\varepsilon(\infty)_{relax}$ or both. A previous DR spectroscopic study that investigated the temperature dependence of the dielectric relaxation of water showed that $\Delta\varepsilon_{\gamma_2}$ decreases with increasing temperature by expanding the frequency range to 0.4 THz^{38,42}. The result may depend on the measurement range and fitting model used due to its small abundance to bulk water $\Delta\varepsilon_{\gamma_1}$; however, a similar result has been obtained by using THz time-domain spectroscopy (THz-TDS) to analyze the relaxational and vibrational modes of water in the THz region⁴². These findings suggest that the origin of the temperature-dependent decrease in $\varepsilon_{\gamma_1}(\infty)$ observed in pure water can be approximated by $\Delta\varepsilon_{\gamma_2}$. By contrast, the $\Delta\varepsilon_{\gamma_2}$ observed in the lysozyme solution had a smaller temperature dependence than that of pure water (**Figure 2**), and therefore, the relaxation origin may not be the same as that observed in pure water.

To evaluate $\Delta\varepsilon_{\gamma_2}$ we next used THz-TDS, which allows direct comparison of dielectric spectra between the lysozyme solution and pure water in the THz region (0.3–2.5 THz). As we have shown that the real part (ε') of dielectric permittivity obtained using this method has a larger measurement error than the imaginary part (ε'')³⁹, we used only ε'' for the analysis (see Figure S6 for ε'). The lysozyme concentration was increased to 28.6 wt% for sensitively detecting any lysozyme-derived changes in relaxation modes. To evaluate ε'' only derived from relaxation modes of water that interact with lysozyme, the lysozyme-derived spectrum, which has been assigned as underdamped vibrational modes by Yamamoto et al.¹⁵ (Figure S6A), was subtracted from the measured spectrum (Figure S6B), and it was further normalized by the fraction of water. Obtained ε'' spectra of water in the lysozyme solution were

compared with that of pure water at different temperatures (**Figure 3A**). These difference spectra revealed that the presence of lysozyme increased ε'' of the lysozyme-interacting water in the THz region that likely included the frequency of γ_2 relaxation (**Figure 3B**), consistent with the prediction from the microwave DR measurement described above. Moreover, in the range of 20–35 °C, the ε'' overlapping with γ_2 peak (0.3–1 THz) tended to increase rather than decrease as the sample temperature increased (**Figure 3**), which was opposite to the temperature dependence of $\Delta\varepsilon_{\gamma_2}$ for pure water reported previously³⁸. This result suggests that the origin of $\Delta\varepsilon_{\gamma_2}$ observed in the presence of lysozyme is different from that characterized in pure water, although the detected temperature-dependence was at a level close to the measurement error (**Figure 3**).

Taken together, the combined microwave DR and THz-TDS measurements indicate that the incident 0.1-THz radiation selectively perturbed water dynamics with increased mobility (i.e., fewer H-bonds) due to interaction with lysozyme. The result of THz-TDS experiment also verified that our analysis method using the DR measurements below 14 GHz frequency, can approximately predict dielectric properties, including those in the THz region.

Fig. 3. Dielectric spectral analysis for the imaginary part of THz-TDS measurements. (A) Solid lines represent spectra of water in the 28.6 wt% lysozyme solution at 20 °C (purple), 25 °C (blue), 30 °C (green), and 35 °C (red). The spectra for dehydrated lysozyme¹⁵ were subtracted from the raw spectra and were normalized by the fraction of water (= 0.714). Dashed lines represent the corresponding spectra of pure water. (B) Subtracting the spectra of pure water (dashed lines of panel A) from those of water contained in the lysozyme solution (solid lines of panel A) gives spectra for lysozyme-interacting water. Color code is the same as shown in panel A. (C) Spectra for the lysozyme solution after 0.1-THz irradiation (red), the control sample without irradiation (blue), and pure water (black) at 25 °C. An enlarged view is shown in the inset. (D) Difference spectrum of the irradiated sample (red) subtracted from non-irradiated control (blue). A red line represents γ_2 relaxation mode given by $\varepsilon''_{\gamma_2}(\omega) = \Delta\varepsilon_{\gamma_2}\omega\tau/(1+j\omega^2\tau^2)$, where $\tau = 0.265$ ps and $\Delta\varepsilon_{\gamma_2}$ is arbitrary. All data are shown as means of four measurements. The measurement errors indicated by shading are given as follows. Panel A: $\sigma/0.714$ and σ for water in the lysozyme solution and pure water, respectively, where σ is the standard deviation of the four measurements. Panel C: σ . Panel B and D: $\sqrt{\sigma_A^2 + \sigma_B^2}$, where σ_A and σ_B are standard deviations for each original spectrum before subtracting.

The use of calibration of the lysozyme solution before and after irradiation, may indeed demonstrate an effect similar to what the authors proclaim, however without a strong justification for this extrapolation one cannot accept the conclusion.

We consider that the strong justification for this extrapolation was made by THz-TDS and NMR measurements and the analyses. As mentioned in above our reply, (i) the irradiation-dependent changes in $\Delta\varepsilon_{\gamma_2}$ that were predicted from the DR analysis agree well with the direct measurements of the THz frequency range, including γ_2 relaxation by THz-TDS (**Figs 2 and 3**). (ii) The irradiation history interpreted based on the dielectric permittivity of water was consistent with that interpreted from the change in methyl-group signal intensity on the lysozyme side measured via NMR (**Fig. 6**).

Another justification is that 0.1 THz irradiation is not equivalent to heating for the change in extrapolated value, in the two different concentrations of the lysozyme solutions (**Fig. 2 and Fig. S4**). This cannot be explained as an artifact of the extrapolation that we did in the analysis. In this respect, we also verified that no interference occurred between the 0.1-THz field and the VNA-generated field, which shows that the extrapolation-based interpretation is unrelated to the interference with the incident 0.1-THz field (please see Materials Methods). In particular, as the VNA-generated duration is similar to the pulse width of incident 0.1 THz wave (~1 μ s), if interference between the two EM fields occurs during 0.1-THz irradiation, there should be a large noise in the obtained dielectric spectrum at each measurement frequency point. However, no such noise was detected in our five measurements per sample (**Fig. S10**).

References

1. Tokunaga, Y. et al. Nonthermal excitation effects mediated by sub-terahertz radiation on hydrogen exchange in ubiquitin. *Biophysical Journal* 120, 2386–2393 (2021).
2. Fröhlich, H. Long-range coherence and energy storage in biological systems. *International Journal of Quantum Chemistry* 2, 641–649 (1968).
3. Ellison, W. J. Permittivity of Pure Water, at Standard Atmospheric Pressure, over the Frequency Range 0–25 THz and the Temperature Range 0–100 °C. *Journal of Physical and Chemical Reference Data* 36, 1–18 (2007).
4. Cametti, C., Marchetti, S., Gambi, C. M. C. & Onori, G. Dielectric Relaxation Spectroscopy of Lysozyme Aqueous Solutions: Analysis of the δ -Dispersion and the Contribution of the Hydration Water. *J. Phys. Chem. B* 115, 7144–7153 (2011).
5. Levy, E., Puzenko, A., Kaatze, U., Ben Ishai, P. & Feldman, Y. Dielectric spectra broadening as the signature of dipole-matrix interaction. II. Water in ionic solutions. *The Journal of Chemical Physics* 136, 114503-114503–6 (2012).

REVIEWERS' COMMENTS

Reviewer #1 (Remarks to the Author):

The revised manuscript has included additional experimental techniques, namely THz-TDS and NMR spectroscopies, in order to attempt to address the concerns raised by the original referees. The authors should be commended on performing such careful experimental work, I am sure the experimental data is of very high quality. While I am personally still a little skeptical of the conclusions the authors have drawn, I think they have done a good job of utilizing this array of techniques to come up with a convincing story, and I think that it deserves publication so that the fundamental atomic-level description the authors generate can be properly debated by others in the field.

Reviewer #3 (Remarks to the Author):

The Authors have satisfied my initial concerns over the article, especially with the inclusion of new experimental results, including an extended dielectric measurement up to 40 GHz.

Please note that the comments are in blue, and the responses are in black.

REVIEWERS' COMMENTS

Reviewer #1 (Remarks to the Author):

The revised manuscript has included additional experimental techniques, namely THz-TDS and NMR spectroscopies, in order to attempt to address the concerns raised by the original referees. The authors should be commended on performing such careful experimental work, I am sure the experimental data is of very high quality. While I am personally still a little skeptical of the conclusions the authors have drawn, I think they have done a good job of utilizing this array of techniques to come up with a convincing story, and I think that it deserves publication so that the fundamental atomic-level description the authors generate can be properly debated by others in the field.

We are again grateful to Reviewer #1 for carefully reviewing our manuscript multiple times and for understanding our careful experimental work and the quality.

From the above comments, we believe that a concern of Reviewer #1 is only the part that “While I am personally still a little skeptical of the conclusions the authors have drawn”. In order to address it, we rewrote the sentence in the beginning of the discussion of a possible mechanistic insight into the sub-THz-irradiation effect on hydration as follows (lines 477-479):

“Based on careful consideration of the results, the most plausible explanation for the sub-THz-irradiation effect on hydration at present is shown in Fig. 7.”

In addition, in the last part of the corresponding discussion, we clarified that future study is needed to provide evidence for the inference (lines 521-526):

“To obtain evidence for this inference, future study should directly observe detailed changes in the water H-bond network throughout the protein surface, including the hydrophobic cavity, before and after the irradiation.”

We believe this is the best response we can provide at this stage, as the concern is a kind personal opinion rather than pointing out a specific remaining problem with our manuscript.

Please also note that our interpretation of the sub-THz irradiation effect on hydration is based not only on the present findings, but also on many previous spectroscopic studies (microwave DR, THz, Raman), Refs 44-50 (lines 479-504).

Furthermore, we revised Figure 7 and the legend to clarify the possibility that, in the hydrophobic

cavity, water molecules originally might have existed in isolation. In the original manuscript, we mentioned this possibility in the main text (lines 508-509 in the revised manuscript) but did not show it in the corresponding figure. We believe that this response has allowed for a more objective description of possible considerations in the initial hydration state (before irradiation) of the protein.

Fig. 7: After revision

Fig. 7: before revision

Reviewer #3 (Remarks to the Author):

The Authors have satisfied my initial concerns over the article, especially with the inclusion of new experimental results, including an extended dielectric measurement up to 40 GHz.

We are again grateful to Reviewer #3 for carefully reviewing our manuscript multiple times and for understanding our careful responses to the comments on the initially submitted manuscript. From the above comments, we believe that Reviewer #3's initial concerns have been addressed and therefore there are no concerns that need to be addressed in this revision.